# Sphingosine 1-phosphate mediates adiponectin receptor signaling essential for lipid homeostasis and embryogenesis

Mario Ruiz [1] ✉, Ranjan Devkota[1], Dimitra Panagaki[1], Per-Olof Bergh [2], Delaney Kaper [1], Marcus Henricsson [2], Ali Nik[1], Kasparas Petkevicius [3], Johanna L. Höög [1], Mohammad Bohlooly-Y[3], Peter Carlsson[1], Jan Borén [2] & Marc Pilon [1] ✉

Cells and organisms require proper membrane composition to function and develop. Phospholipids are the major component of membranes and are primarily acquired through the diet. Given great variability in diet composition, cells must be able to deploy mechanisms that correct deviations from optimal membrane composition and properties. Here, using lipidomics and unbiased proteomics, we found that the embryonic lethality in mice lacking the fluidity regulators Adiponectin Receptors 1 and 2 (AdipoR1/2) is associated with aberrant high saturation of the membrane phospholipids. Using mouse embryonic fibroblasts (MEFs) derived from AdipoR1/2-KO embryos, human cell lines and the model organism *C. elegans* we found that, mechanistically, AdipoR1/2-derived sphingosine 1-phosphate (S1P) signals in parallel through S1PR3-SREBP1 and PPARγ to sustain the expression of the fatty acid desaturase SCD and maintain membrane properties. Thus, our work identifies an evolutionary conserved pathway by which cells and organisms achieve membrane homeostasis and adapt to a variable environment.

Lipid membranes function as barriers that not only delimit cell boundaries, but also define internal compartments within eukaryotic cells. Approximately one third of all proteins are integral membrane proteins[1] and many more are membrane-associated proteins. Therefore, multiple cellular processes occur at or in connection with membranes. Perturbations in membrane composition and properties can have dramatic consequences for the cell/organism. For instance, lipotoxicity can often be explained by excessive membrane-lipid packing that in turn can be caused by a diet rich in saturated fats or by the loss of membrane regulators such as the Adiponectin Receptors (AdipoR1 and AdipoR2)[2–5].

The AdipoR1/2 proteins are often discussed in the context of diabetes due to their ability to promote insulin sensitivity and oppose hepatic steatosis[6–8]. AdipoR1 and AdipoR2 single knockout mice are viable but the simultaneous ablation of both AdipoRs (DKO) causes embryonic lethality[9], though the primary cause of death has not been investigated. Studies in mammalian cells of different origin and in the nematode *C. elegans* have shown that a primary and evolutionary conserved function of the AdipoR1/2 proteins is to promote the expression of fatty acid desaturases to maintain membrane homeostasis[10–12]. In particular, an RNAseq study highlighted that stearoyl-CoA desaturase (*SCD*) was the most robustly downregulated gene in cells lacking AdipoR2[2]. At the molecular level, AdipoR1/2 possess an intrinsic ceramidase activity that catalyzes the hydrolysis of ceramides to produce sphingosine and free fatty acids[13]. The resulting sphingosine can be phosphorylated by sphingosine kinases 1 and 2 (Sphk1/2) to produce the signaling molecule sphingosine 1-phosphate (S1P) and, speculatively, mediate AdipoR1/2 function[14–16].

[1]Dept.Chemistry and Molecular Biology, Univ. Gothenburg, 405 30 Gothenburg, Sweden. [2]Dept. Molecular and Clinical Medicine/Wallenberg Laboratory, Institute of Medicine, Univ. of Gothenburg, 414 67 Gothenburg, Sweden. [3]Discovery Sciences, BioPharmaceuticals R&D, AstraZeneca, Gothenburg, Sweden. ✉ e-mail: mario.ruiz@gu.se; marc.pilon@cmb.gu.se

S1P is an essential[17] and bioactive lipid involved in several cellular and physiological processes including cell survival, migration, inflammation and development[18–20]. S1P can be exported out of the cells and signal through a family of G-protein-coupled receptors, GPCRs (S1P receptors 1–5, S1PR1-5)[21]. Alternatively, S1P can also bind to several intracellular targets such as the peroxisome-proliferator activated receptor γ (PPARγ) and the histone deacetylases 1 and 2 (HDAC1/2) to modulate the transcriptional response of the cell[22,23].

Here, to better understand the cause of AdipoR1/2 embryonal lethality, we generated DKO mouse embryos and explored the hypothesis that deficient S1P signaling may have led to a failure in membrane homeostasis. We found that AdipoR1/2-derived S1P mediates membrane homeostasis by activating the sterol regulatory element-binding protein-1 (SREBP1) via S1PR3 and, separately, PPARγ.

## Results

### AdipoR1/2 DKO lethality and excess phospholipid saturation

Deficiency of both AdipoR1 and AdipoR2 leads to embryonic lethality in mice[9], but the underlying cause is not well understood. The recent emergence of AdipoR1/2 as regulators of membrane composition and fluidity[12] suggests that membrane composition defects could contribute to the DKO embryonic lethality. To test this hypothesis, we first crossed AdipoR1/2 double heterozygous knockout mice with each other and collected E12.5 and E15.5 embryos (Fig. 1A). A single DKO embryo was found at E15.5 and showed obvious morphological defects, drastic reduction in size and was pale in color (Fig. 1A and Supplementary Data 1). The extremely low frequency of DKO embryos isolated is noteworthy: we expected to find one DKO embryo out of every sixteen embryos (mendelian ratio), but instead found only one in forty-six. In contrast, we recovered the expected proportion of DKO embryos at E12.5 (Fig. 1A and Supplementary Data 1), indicating that DKO embryos are formed, but failed and were resorbed between the E12.5 and E15.5 stages. To gain insight into the causes of AdipoR1/2 embryonic lethality, we performed histology, lipidomics and proteomics of E12.5 embryos (Fig. 1B). Histology of E12.5 wild-type (WT) and DKO embryos did not reveal any major morphological defects in the DKO embryos (Fig. 1C and Supplementary Fig. 1A–H) suggesting that morphological defects are preceded by molecular changes.

Lipidomics analysis of multiple lipid classes allowed for a clear separation of WT, AdipoR1-KO, AdipoR2-KO and DKO E12.5 embryos in a principal component analysis (PCA) (Fig. 1D). The PCA further revealed that most of the separation between genotypes (PC1 accounted for 78% of the variance) is explained by differences in the abundance of palmitic acid (PA, 16:0) in membrane phospholipids (Supplementary Fig. 1I). A volcano plot comparing WT and DKO embryos revealed significant changes in 56 lipid species/classes, most of them being phospholipids (Fig. 1E). Phosphatidylcholines and phosphatidylethanolamines (PC and PE respectively) species containing saturated fatty acids (SFA), especially PA, were strikingly over-represented in DKO embryos whereas monounsaturated fatty acids (MUFA; e.g., oleic acid (OA, 18:1) and palmitoleic (PalOA, 16:1)) and polyunsaturated fatty acids (PUFA; e.g., arachidonic acid (AA, 20:4)) were more abundant in the phospholipids of WT embryos (Fig. 1E). Accordingly, a heat map of fatty acid species in several lipid classes in all four genotypes shows a clear gradient in phospholipid saturation:  DKO > AdipoR2-KO > AdipoR1-KO > WT  (Supplementary Fig. S1J and Supplementary Data 2). In contrast, we did not find any differences in the ratios between PC and several other lipids classes, namely PE, triacylglycerols (TAG), free cholesterol (FC), sphingomyelin (SM), ceramide (Cer), dihydroceramides (DiCer), lactosylceramides (LacCer) or glucosylceramides (GluCer) (Supplementary Data 2). These results are consistent with AdipoR1/2 being essential proteins that sustain fatty acid desaturation to regulate membrane composition/fluidity. Moreover, the lipidomics analysis indicates a

more prominent role for AdipoR2 over AdipoR1, which is consistent with previous studies human cells[10,24].

To further investigate the cause of the lethality of DKO embryos, we compared the proteomes of WT and DKO E12.5 embryos and identified 7093 proteins, including 266 that were misregulated in the mutant: 185 proteins were upregulated and 81 downregulated in DKO embryos (Fig. 1F and Supplementary Data 3). To find associations among the 266 misregulated proteins, we performed a gene set enrichment analysis (GSEA), which grouped 97 protein into 16 pathways that differed significantly between the two genotypes. Proteins from multiple metabolic pathways, including glycerolipid metabolism and fatty acid metabolism (Fig. 1G, H, Supplementary Fig. 1K and Supplementary Data 3), were present at higher levels in the DKO embryos, as indicated by a positive enrichment score. In contrast, the "SNARE interactions in vesicular transport" pathway was the only one with a significant negative score based on protein abundance (Fig. 1H, Supplementary Fig. 1K and Supplementary Data 3), which is again consistent with membrane defects in the DKO embryos.

AdipoR2 regulates cell membrane fluidity cell non-autonomously[25] and, interestingly, our proteomics analysis also revealed that the abundance of several apolipoproteins, that can mediate lipid transport between cells and tissues, was diminished in the DKO embryos. More specifically, Apolipoprotein A1 (ApoA1) and Apolipoprotein B100 (ApoB), which are the major components of high-density lipoproteins (HDL) and low-density lipoproteins (LDL) respectively, were reduced to ~50% of the WT levels (Fig. 1I). This result is consistent with poor lipid exchange between tissues[25]. Similarly, the levels of Apolipoprotein E (ApoE) and Apolipoprotein M (ApoM) were also about half of WT levels (Fig. 1J). ApoM is particularly interesting because its ligand, S1P, has been suggested to mediate AdipoR1/2 signaling[14]. Indeed, the excessive saturation of PC in AdipoR2 KO embryonic brains at E15.5 (Fig. 1K and Supplementary Data 2) was accompanied by a significant reduction of S1P and its precursor sphingosine (Sph) (Fig. 1L and Supplementary Fig. 1L). Note that the frequency of AdipoR1/2 DKO embryos at E15.5 is very low and S1P was below detection in the single AdipoR1/2 DKO embryo isolated (Fig. 1A, L).

Collectively, our results suggest that AdipoR1/2 DKO embryonic lethality may be caused by metabolic problems that are secondary consequences of membrane malfunction due to an excess of SFA-containing phospholipids.

### DKO MEFs reproduce the membrane defects of DKO embryos

We next sought to decipher the precise mechanism by which AdipoR1/2 regulates phospholipid saturation and hence membrane homeostasis. For that, we moved to a more flexible experimental system and obtained mouse embryonic fibroblasts (MEFs) from WT, AdipoR1-KO, AdipoR2-KO and DKO embryos at E12.5 (Fig. 1B) and studied their membranes. DKO MEFs showed membrane composition defects similar to those observed in embryos, and the addition of PA to the culture exacerbated these defects. More specifically, DKO MEFs accumulated higher amounts of PA in PC and PE at the expense of OA and AA (Fig. 2A, B, Supplementary Fig. 2A–C and Supplementary Data 4). Importantly, the supplementation of the MUFA OA partially corrected the defects caused by PA treatment (Fig. 2A, B, Supplementary Fig. 2A, B and Supplementary Data 4). As in embryos, phospholipids in DKO MEFs were more saturated than in the single AdipoR2-KO (Fig. 2A, B, Supplementary Fig. 2A–C and Supplementary Data 4) and phospholipids in AdipoR2-KO were more saturated than in AdipoR1-KO (Supplementary Fig. 2D, E). These data are consistent with previous publications showing that AdipoR1/2 deficient cells lack the ability to desaturate fatty acids[2,10]. Given that the AdipoR1-KO had a less prominent phenotype in both cells and embryos, our later studies mostly focused on AdipoR2-KO and DKO cells.

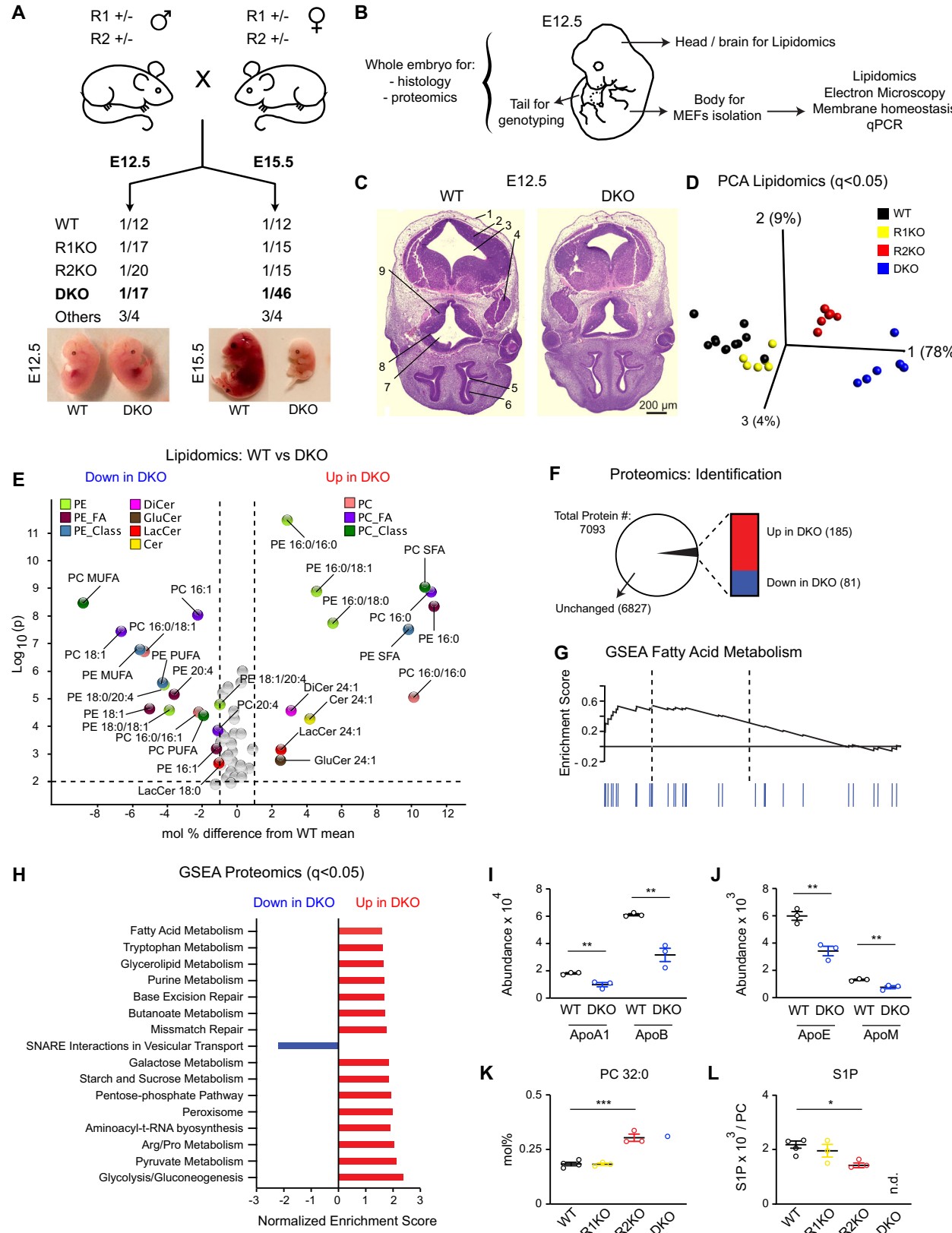

Excess of SFA in phospholipids leads to increased lipid packing, loss of membrane fluidity and, secondarily, affects several physiological processes[26,27]. To confirm the excessive membrane packing in AdipoR2-KO and DKO cells, we imaged live MEFs stained with Laurdan dye and calculated their generalized polarization (GP) index (Fig. 2C). Interestingly, AdipoR2-KO and DKO MEFs showed little differences

compared to WT MEFs under basal conditions, but their GP indexes dramatically increased when challenged with PA in a concentration-dependent manner, indicating extreme lipid packing (Fig. 2D–G and Supplementary Fig. 2F). As in the embryos, DKO MEFs showed an aggravated phenotype and their membranes were distinctly more packed than those of AdipoR2-KO MEFs (Fig. 2D–G). Strikingly long

**Fig. 1 | Membrane-lipid composition defects precede embryonic lethality in DKO mice. A** Mouse crosses and genotype frequencies at E12.5 ($n = 218$) and E15.5 ($n = 46$). See also Supplementary Data 1. **B** E12.5 embryo showing parts used. **C** Sections of WT and DKO embryos at E12.5 stained with H&E. (1) surface ectoderm; (2) fourth ventricle; (3) basal plate of metencephalon; (4) trigeminal (V) ganglion; (5) lumen of primitive nasal cavity; (6) olfactory epithelium; (7) third ventricle; (8) optic recess; (9) region of optic chiasma. More sections in Supplementary Fig. 1A–H. **D** Lipidomics-based PCA of E12.5 heads. Quantities analyzed: mol% of fatty acids species and type (SFA, MUFA, PUFA), and total amount of PE, Cer, DiCer, GlcCer, LacCer, TAG and FC normalized to total PC. Variables used are shown in Supplementary Fig. 1I and Supplementary Data 2. **E** Volcano plot of lipid species differing between WT and DKO at E12.5, with a $p < 0.01$ based on t-tests (two sided and assuming normality). Species differing by ≥1% mol are labeled; species measured are in Supplementary Data 2. **F** Number of proteins identified in WT and DKO embryos at E12.5 and with altered levels ($p < 0.01$). See Data S3. **G** GSEA Enrichment plot of the KEGG Fatty Acid Metabolism Pathway. **H** Gene Set Enrichment Analysis (GSEA) of proteomics data: 16 KEGG pathways are over-represented in DKO embryos at E12.5. See Supplementary Fig. 1K and Supplementary Data 3. **I, J** Abundance of apolipo-proteins in WT and DKO embryos at E12.5. $n = 3$ biologically independent replicates per condition. **K, L** PC 32:0 and S1P abundance in brains at E15.5. $n = 4, 3, 3$ and 1 biologically independent replicate for WT, R1KO, R2KO and DKO, respectively. Lipidomics in Supplementary Fig. 1L and Supplementary Data 2. (n.d. non-detected) **I–L** shows mean ± SEM and t-tests (two sided and assuming normality) were used to identify significant differences between treatments. **$p < 0.01$ and *$p < 0.05$. See Supplementary Fig. 1 and Supplementary Data 1–3. Source data provided as a Source Data file.

and thin solid-like structures exclusively appeared in AdipoR2-KO and DKO MEFs after PA treatment. We used Bodipy-C12 to confirm the lipidic nature of these abnormal structures (Fig. 2H) and, given that phospholipids rich in SFA pack tighter and can form solid-like domains in the endoplasmic reticulum (ER)[28], we investigated whether the structures observed in AdipoR2 deficient MEFs corresponded to ER membranes. For that we co-stained live MEFs with ER-Tracker and Laurdan or Bodipy-C12 and found that the long and thin solid-like structures in the cytoplasm of AdipoR2-KO MEFs correspond indeed to ER membranes (Fig. 2I and Supplementary Fig. 2G). Next, we built 2.5D image-reconstructions containing spatial information of the solid-like ER structures in AdipoR2-KO MEFs and found that some were oriented parallel to the culture surface, whereas others stood vertically along the Z-axis (Fig. 2J).

To better characterize the membranes in MEFs derived from DKO embryos, we used transmission electron microscopy to image WT and DKO MEFs under basal condition or following PA treatment to challenge the membranes (Supplementary Fig. 2H–K). Electron micrographs of DKO MEFs presented unique spiral membranes in the cytoplasm that were totally absent in WT MEFs, both under basal conditions and upon PA treatment (Fig. 2K and Supplementary Fig. 2L, M). The DKO MEFs also had closely apposed membranes in the cytoplasm that increased seven-fold upon PA treatment (Supplementary Fig. 2N–P). Additionally, nuclear envelope budding events, which have been previously identified in cells under several stress conditions[29], were also present in most of the DKO MEFs upon PA treatment (Fig. 2L and Supplementary Fig. 2Q, R). These membrane defects are likely ER abnormalities given the fluorescent microscopy study described earlier (Fig. 2E–J). Note however that DKO MEFs preserved ER-mitochondria contact points (Supplementary Fig. 2S–V) and the ability to make lipid droplets (Supplementary Fig. 3A–C) in basal conditions and when treated with PA or OA.

Finally, and as in the brains of the AdipoR2-KO embryos at E15.5, S1P and its precursor Sph were reduced in DKO MEFs (Fig. 2M, N and Supplementary Data 4) under basal conditions, or after PA treatment.

Overall, these experiments demonstrate that MEFs are a useful model to recreate and study DKO embryonal membrane defects. We next focused on elucidating the molecular mechanisms/pathways by which AdipoR1/2 contribute to membrane homeostasis regulation.

## AdipoR1/2 DKO membrane defects are rescued by S1P

AdipoR1/2 possess an intrinsic ceramidase activity in vitro that can catalyze the hydrolysis of ceramides to produce sphingosine and free fatty acids[13]. The resulting sphingosine can be phosphorylated to produce the signaling molecule S1P. Consistent with the Adi-poR1/2 ceramidase activity, total ceramides are increased in DKO MEFs[14], whereas S1P and Sph levels were reduced (Fig. 2M, N and ref. 14). On the other hand, over-expression of AdipoR2 in the liver resulted in a significant increase of S1P levels[6]. Therefore, to follow the hypothesized connection between AdipoR1/2 and S1P, we investigated whether S1P mediates AdipoR1/2 regulation of

membrane homeostasis. For that, we challenged DKO MEFs with PA in the presence or absence of exogenously supplied S1P and monitored their membrane composition and properties. Lipidomics analysis showed that S1P significantly decreased the amount of PA in PE, whereas it caused an increase in OA and total PUFA in PE (Fig. 2O–R and Supplementary Data 4). Second, Laurdan dye assays confirmed that membranes are less packed in DKO cells treated with S1P (Fig. 2S). Further, it is a well-established observation that increased levels of SFA induce ER-stress[2,30] and, in agreement with this, we found that S1P treatment reduced the expression of ER-stress markers Ddit3 and Atf4 (Fig. 2T, U and Supplementary Data 5).

To further verify our observation that S1P promotes membrane homeostasis, we silenced AdipoR2 expression in HEK293 cells (Supplementary Fig. 3D) then challenged these cells with PA in the presence or absence of S1P. As in MEFs, S1P diminished the saturation levels in membrane phospholipids. More specifically, S1P reduced PA in PC and PE and increased several unsaturated fatty acids: OA in PC and PE, eicosadienoic acid (20:2 n-6) in PC and PalOA in PE (Supplementary Fig. 3E–J and Supplementary Data 6). Adi-poR2 silencing in the presence of exogenous palmitate abnormally activates the de novo ceramide synthesis pathway, leading to increased levels of DiCer[31]. Here, we found that S1P partially normalizes the levels of DiCer, SM, GlcCer and LacCer (Supplementary Fig. 3K–O and Supplementary Data 6). Consistently, S1P also improved membrane packing, as indicated by a lower GP index (Supplementary Fig. 3P, Q). Moreover, the addition of a fluidizing MUFA, OA, fully compensated the excessive lipid packing caused by PA in AdipoR2-siRNA treated cells (Supplementary Fig. 3Q).

Because orthologs of the AdipoR2 protein are conserved in eukaryotes[12,32], we next asked whether S1P protection against membrane-rigidifying conditions is evolutionary conserved. A characteristic phenotype of the *C. elegans paqr-2* mutants (*paqr-2* encodes an AdipoR2 ortholog) is their inability to perform homeoviscous adaptation and grow at 15 °C (note that wild-type N2 worms are able to grow between 15 and 25 °C)[32]. The *paqr-2* mutants, which arrested as L1 larvae at 15 °C on standard media, significantly benefitted from the addition of S1P to the culture plate, which allowed them to grow to the L3-L4 stages (Supplementary Fig. 3R–T). Additionally, the *paqr-2* mutants exhibit a characteristic withered morphology of the thin membranous tail tip[32] that was also partially corrected by S1P supplementation (Supplementary Fig. 3U, V). As a consequence of the excess of SFA in membrane phospholipids, *paqr-2* mutants also produce an abnormally small brood size accompanied by defects in the germ line[25,32]. Remarkably, injection of S1P directly into the gonad of *paqr-2* mutant worms partially suppressed their reproduction defects and actually led to a doubling of their brood size (Supplementary Fig. 3W, X).

Taken together, we conclude that S1P promotes membrane homeostasis and compensates for the absence of the evolutionary conserved membrane fluidity regulator AdipoR2/PAQR-2.

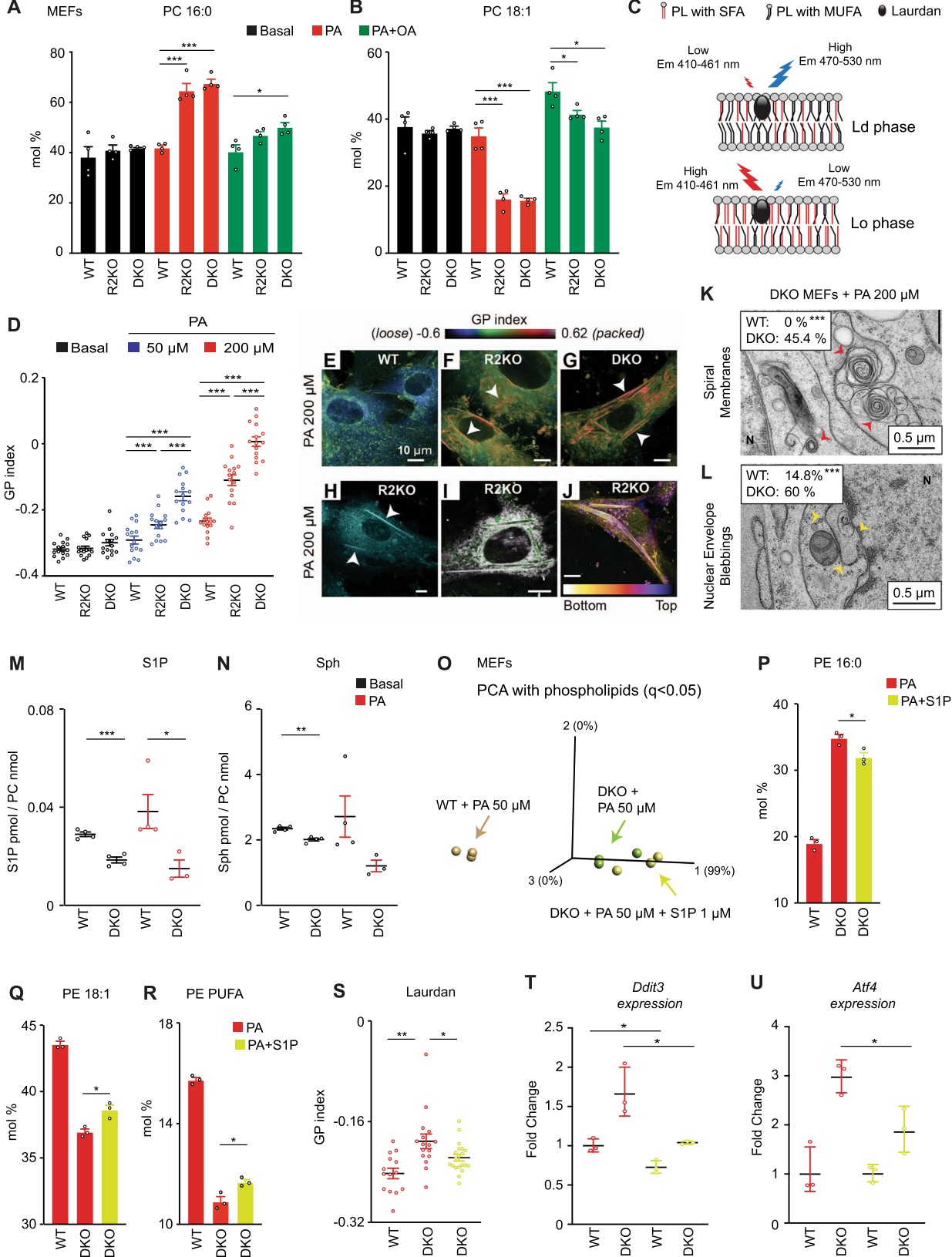

### Exogenous S1P promotes membrane health in PA-overwhelmed WT cells

Small amounts of PA are particularly toxic to cells/organism lacking AdipoR1/2[10,24]. Even though WT cells react to high doses of PA by increasing S1P and Sph production (Fig. 3A and Supplementary Data 4), WT cells are also sensitive to high concentrations of PA and

suffer membrane rigidification and death[14,33]. Therefore, we asked whether exogenous S1P can increase the resistance of WT cells against excessive amounts of PA. To address this question, we first used WT MEFs and HEK293 cells, treated them with PA 400 µM in the presence or absence of S1P and stained them with Laurdan dye. We found that exogenous S1P protected MEFs (Fig. 3B, C) and HEK293 cells

**Fig. 2 | S1P rescues membranes in AdipoR2 KO MEFs. A, B** PA (16:0) and OA (18:1) abundance in PC of MEFs ± PA 200 μM ± OA 200 μM for 18 h. $n = 4$ biologically independent replicates per condition. Additional lipidomics in Supplementary Fig. 2A–C and Supplementary Data 4. **C** Laurdan dye method. **D** GP index of Laurdan-stained MEFs. $n = 15$ images, except for R2KO on basal media where $n = 14$. **E–G** Pseudocolor images from D. Packed structures (white arrows). Enlarged examples in Supplementary Fig. 2F. **H** Confocal image of R2KO MEFs stained with C1-Bodipy-C12. **I** R2KO MEFs stained with Laurdan (green) and ER-tracker (purple); merged signals in white. Originals in Supplementary Fig. 2G. **J** Color-coded image of Z-stack of R2KO MEFs stained with C1-Bodipy-C12. **K, L** Electron microscopy of spiral membranes (red arrows) and nuclear envelope blebbing (yellow arrows). $n = 27$ and 22 sections for WT and DKO, respectively. "N" indicates nuclei. Larger images and quantifications in Supplementary Fig. 2L, M, Q, R. **M, N** S1P and Sph abundance in MEFs ± PA 200 μM for 6 h. $n = 4$ biologically independent replicates except for DKO + PA where $n = 3$. From Supplementary Data 4. **O** PCA based on varying fatty acid species in PC and PE of MEFs. **P– R** PA, OA and PUFA levels in PE of MEFs + PA 50 μM ± S1P 1 μM. $n = 3$ biologically independent replicates per condition. From Supplementary Data 4. **S** GP index of MEFs + PA 50 μM ± S1P 1 μM and Laurdan-stained. $n = 14, 16$ and 20 separate images for WT, DKO and DKO + S1P. **T, U** qPCR in MEFs + PA 50 μM ± S1P 10 μM. $n = 3$ biologically independent replicates per condition. Data shows mean ± SEM, except ±SD in **T, U**. $t$-tests (two sided and assuming normality) identified significant differences between treatments. $*p < 0.05$, $**p < 0.01$, $***p < 0.001$. See Supplementary Figs. 2, 3 and Supplementary Data 4, 5. Source data in Source Data file.

(Supplementary Fig. 4A and Supplementary Data 5) from PA-induced membrane packing, as indicated by the lower GP index. Note that even though S1P and OA gave a similar overall GP index in MEFs, both treatments were clearly distinguishable: whereas PA + S1P looked similar to vehicle treated cells, PA + OA treated MEFs had very loosely packed membranes (colored in dark blue, indicating low GP index) and distinct small areas with high GP index (colored in red), which probably were lipid droplets (Fig. 3B). This is consistent with a previous study showing that OA prevents PA-induced apoptosis by channeling PA into TAG[33].

We also performed fluorescence recovery after photobleaching (FRAP) experiments in live HEK293 cells to assess membrane properties more dynamically. Consistently, S1P improved membrane fluidity in PA treated cells (Supplementary Fig. 4B, C): as little as 10 nM S1P was enough to significantly promote fluidity in HEK293 cells treated with PA 400 μM (Supplementary Fig. 4D and Supplementary Data 5). An 18:1 fatty acid can be synthesized from PA via PalOA or via stearic acid (18:0) (Supplementary Fig. 4E). Interestingly, WT cells reacted to PA by boosting SCD activity to convert PA into PalOA and hence prioritizing desaturation over elongation (Fig. 3D, E, Supplementary Fig. 4F–I and Supplementary Data 6). In agreement with our model, S1P further promoted desaturation and increased the levels of 16:1 and 18:1 in PC and PE, as determined through lipidomics analysis (Fig. 3D, E and Supplementary Fig. 4F-I and Supplementary Data 6).

Since AdipoR1/2 are ubiquitously expressed, we hypothesized that S1P protection against PA-induced membrane rigidity is not limited to MEFs and HEK293 cells. To test this hypothesis, we used Jurkat E6-1 (human T lymphocyte cell line) and INS-1E (rat pancreatic beta-cell line). Jurkat E6-1 membrane packing, determined using the GP index following Laurdan dye staining, increased in a PA concentration-dependent manner (Supplementary Fig. 4J) and S1P reversed the PA effect (Supplementary Fig. 4K). Similarly, S1P abolished the rigidifying effect of PA in INS-1E cells as measured by FRAP (Supplementary Data S5) and Laurdan dye staining GP index (Fig. 3F). INS-1E cells can be used to monitor insulin secretion, which depends on delicate membrane trafficking and fusion events[34]. PA treatment of INS-1E cells reduced glucose-stimulated insulin secretion (GSIS), which again was partially rescued by the addition of S1P (Fig. 3G). This result is in line with previous results showing that S1P inhibits PA-induced apoptosis in INS-1 cells[14] and in L6 skeletal muscle cells[35].

An alternative way to protect WT cells against overwhelming amounts of PA is by over-expressing AdipoR1/2[31,36]. AdipoR1/2 contain a $Zn^{2+}$ ion inside the catalytic domain which is essential for their ceramidase activity. In particular, the $Zn^{2+}$ ion is coordinated by three evolutionary conserved His residues (illustrated in Fig. 3H) and the ceramidase activity is hampered when these His are mutated[13,37]. Therefore, we used HEK293 cells to over-express wild-type and mutated forms of AdipoR2 in which one of the His (H348) or all three of them (H202, H348 and H350) were changed to Ala and confirmed the expression of the constructs by Western-blot (Fig. 3H, I), and then assessed membrane health. We found that while the wild-type AdipoR2 efficiently protected HEK293 cells against the membrane-rigidifying

effects of PA, the single H348A AdipoR2 mutant and the triple His mutant did not (Fig. 3J). Similarly, we used *paqr-2* mutant worms to over-express WT PAQR-2::GFP or PAQR-2::GFP versions where one or all of the zinc coordinating His were mutated. We first confirmed their expression (Supplementary Fig. 4L) and then challenged their membranes by growing the worms at 15 °C or in presence of glucose (note that glucose is readily converted to SFA by the dietary *E. coli*[24]). In both experiments, the WT PAQR-2::GFP and PAQR-2::GFP H511A single His mutant protected against the membrane-rigidifying challenge while the PAQR-2::GFP H365A, H511A, H515A triple His mutant did not (Supplementary Fig. 4M, N). Similarly, the PAQR-2 triple His mutant did not rescue the deformed tail phenotypes of the *paqr-2* mutant (Supplementary Fig. 4O).

Altogether, our results demonstrate that S1P promotes membrane fluidity and suggest that the conserved AdipoR2 catalytic site, capable of a ceramidase activity that would lead to S1P production[13,14], is essential for membrane homeostasis.

## Sphingosine kinases are required for membrane homeostasis

It has been proposed that the sphingosine kinases (Sphk1 and Sphk2) mediate AdipoR1/2 response[14], though this had not yet been experimentally tested. To begin to understand the role of Sphk1/2 in membrane fluidity, we knocked-down Sphk1 and Sphk2 expression in HEK293 cells (Supplementary Fig. 5A, B) and assessed membrane health. Interestingly, single siRNA against Sphk1 or Sphk2 did not affect membrane packing either in basal conditions or upon PA treatment (Fig. 4A and Supplementary Data 5). However, membrane packing was notably increased after simultaneous knockdown of Sphk1 and Sphk2 (Fig. 4A and Supplementary Data 5). A single sphingosine kinase has been identified so far in *C. elegans* and the corresponding *sphk-1* mutant worms had more rigid membranes as measured by FRAP (Supplementary Fig. 5C–E). Accordingly, the *sphk-1* mutant was sensitive to PA-loaded bacteria: *sphk-1* mutant worms grew slower and showed a tail tip morphology defect similar to that of the *paqr-2* mutant (Supplementary Fig. 5F, G and Supplementary Data 5). Note however that the growth and tail tip phenotypes of the *sphk-1* mutant were not as severe as in the *paqr-2* mutant, suggesting that worms may have a second yet unidentified sphingosine kinase. Importantly, the addition of S1P to the cell culture rescued the phenotypes caused by Sphk1/2 silencing in HEK293 cells (Fig. 4B, C). Altogether, these results are consistent with sphingosine kinase activity being an evolutionary conserved effector of AdipoR1/2 during membrane homeostasis.

## Parallel activation of S1PR3-SREBP1 and PPARγ by S1P

Multiple pathways mediate S1P biological actions. Canonical S1P signaling starts with the activation of one of the five S1P membrane receptors (S1PR1-5)[21], but S1P intracellular targets have also been discovered[22,38] (Fig. 4D, E). We therefore sought to elucidate which pathway(s) are relevant for AdipoR2-S1P membrane homeostasis.

To investigate whether S1PR1-5 regulate membrane properties, we first determined the expression levels of the S1PRs in MEFs. We detected a strong expression of S1PR3, moderate expression of S1PR1-

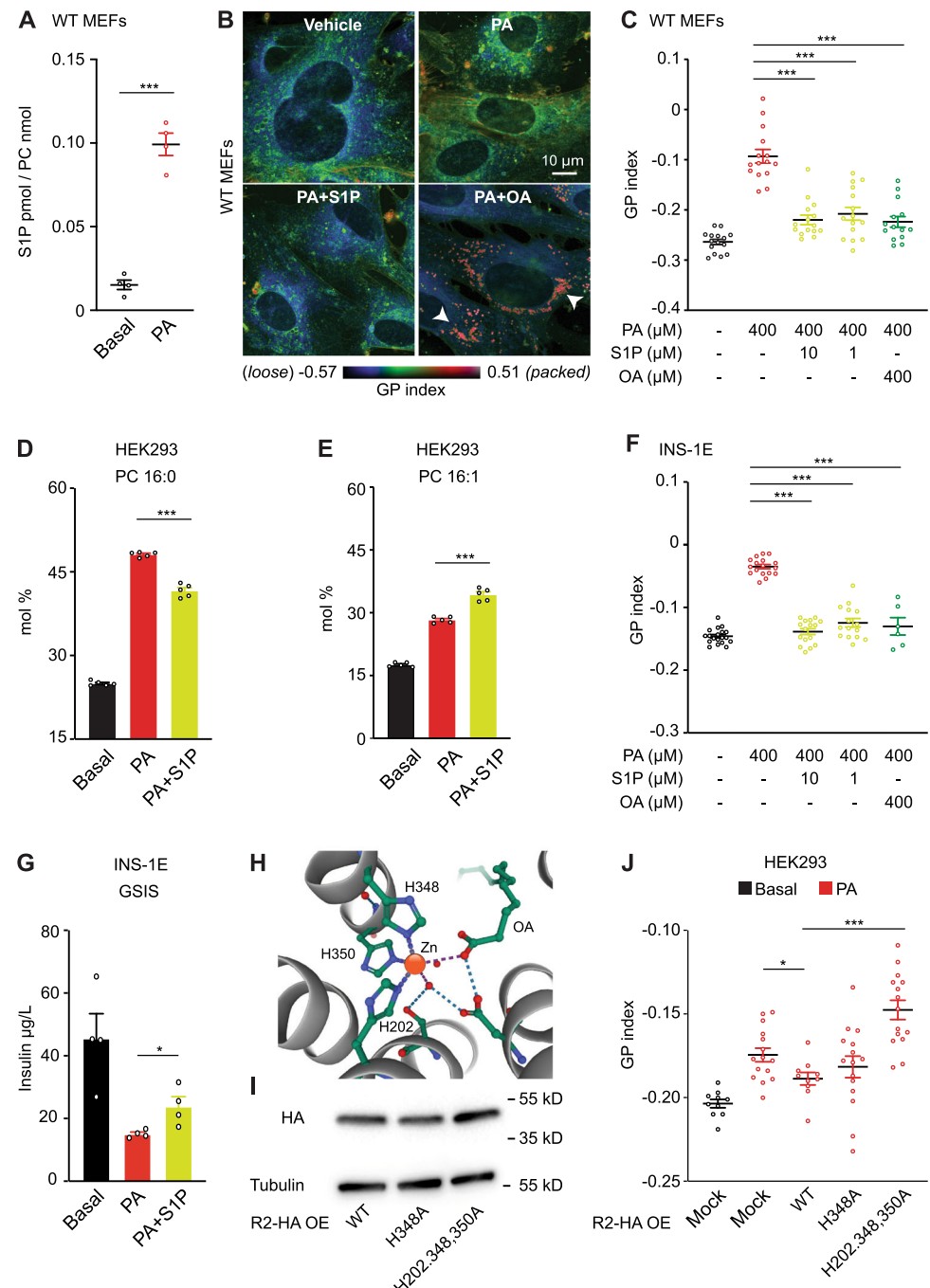

**Fig. 3 | S1P promotes resistance against saturated fatty acids by improving membrane homeostasis in mammalian cells. A** S1P abundance in WT MEFs ± PA 400 μM for 12 h. $n = 4$ biologically independent replicates for each condition. Related lipidomics in Supplementary Data 4. **B**, **C** Pseudocolor images and GP index of WT MEFs ± PA 400 μM ± S1P 10 or 1 μM ± OA 400 μM for 18 h and stained with Laurdan. Note the large number of "small" and highly packed lipid droplets induced by PA + OA (white arrows). In C, $n = 15$ images analyzed per condition, except for PA + OA where $n = 14$. **D**, **E** PA (16:0) and PalOA (16:1) abundance in the PC and PE of HEK293 cells + PA 400 μM ±S1P 10 μM for 24 h. $n = 5$ biologically independent replicates per condition. Related lipidomics are in Data S5. **F** Average GP index of the rat insulinoma INS-1E ± PA 400 μM ± S1P 10 or 1 μM ± OA 400 μM and stained with Laurdan. For each condition, from left to right, $n = 19, 17, 16, 15$, and 6 images analyzed. **G** Glucose-stimulated insulin secretion (GSIS) assay: Amount of insulin secreted (in 2 h) by INS-1E cells pre-exposed ± PA 400 μM ± S1P for 18 h. $n = 4$ biologically independent replicates for each condition. **H** Schematics of the Zn²⁺ atom inside human AdipoR2. PDB 5LX9 entry was used as model[13]. The Zn²⁺ is coordinated by three histidines. **I** Western-Blot of AdipoR2 and Tubulin in HEK293 cells transfected with human AdipoR2: WT sequence, H348A mutant and H202,348,350A mutations. Uncropped blots in Source Data. **J** Average GP index from several images of HEK293 cells transfected with human AdipoR2: WT sequence, H348A mutant and H202,348,350A mutations and challenged with PA 400 μM for 24 h. For each condition, from left to right, $n = 10, 15, 10, 15$ and 15 images analyzed. Data are represented as mean ± SEM. $t$-tests (two sided and assuming normality) were used to identify significant differences between treatments. $*p < 0.05$, $**p < 0.01$, $***p < 0.001$. See also Supplementary Fig. 4 and Supplementary Data 4–6. Source data are provided as a Source Data file.

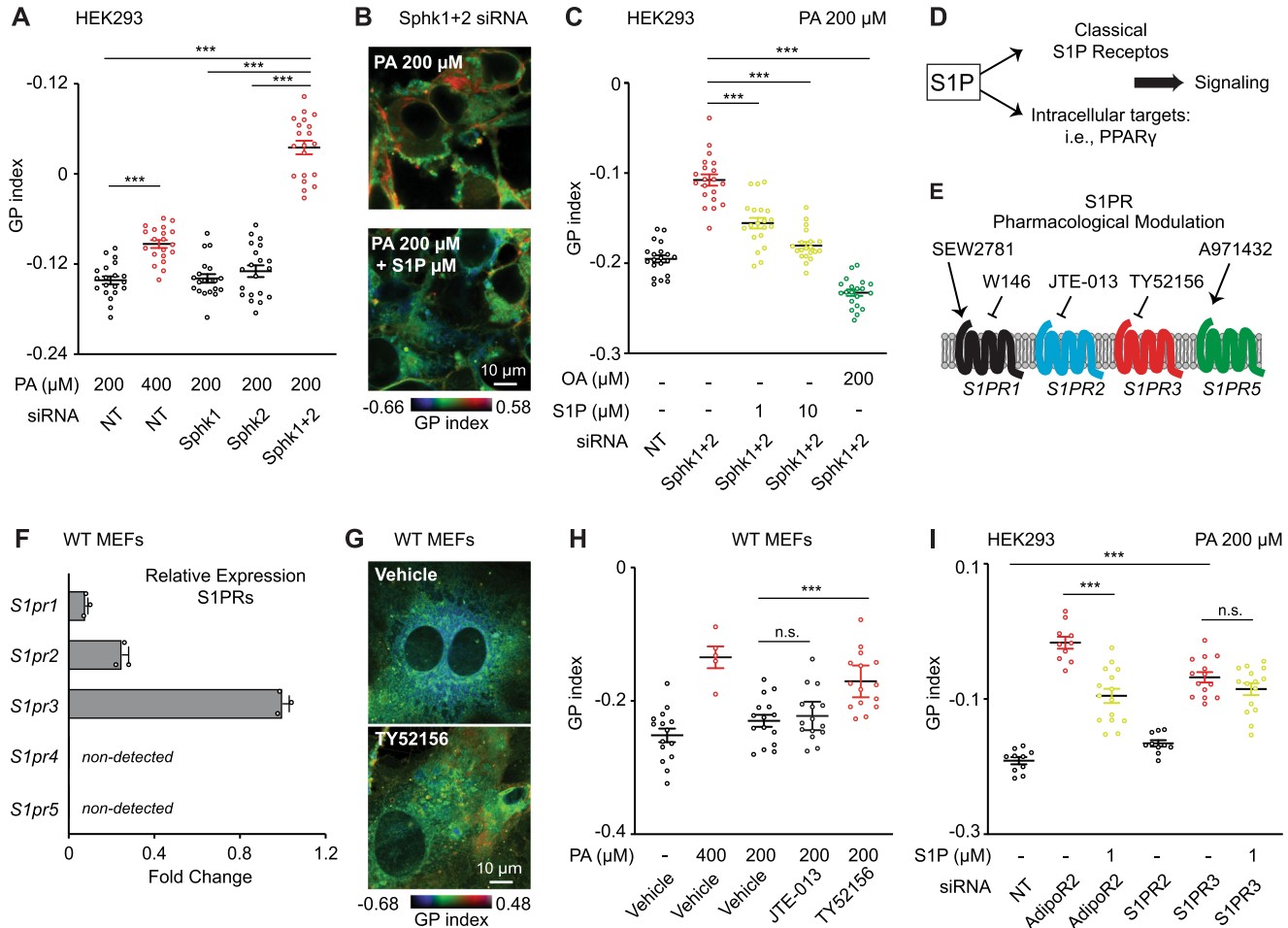

**Fig. 4 | Sphingosine kinases and S1PR3 are required to maintain membrane homeostasis in human/mouse cells. A** Average GP index from several images of NT, Sphk1 and Sphk2 siRNA-treated HEK293 cells challenged with PA 200 μM and stained with the Laurdan dye. *n* = 20 separate images analyzed for each condition. **B, C** Pseudocolor images and average GP index from several images of NT and Sphk1 + 2 siRNA HEK293 cells treated with PA 200 μM ± S1P 1 μM or ± OA 200 μM. In **C**, *n* = 20 separate images analyzed for each condition. **D, E** Schematic representation of S1P signaling and S1PR pharmacological modulators used in this work. **F** Relative expression of S1PRs in WT MEFs measured by qPCR. *n* = 3 technical replicates per condition. **G, H** Pseudocolor images and average GP index from

several images of WT MEFs treated with vehicle, PA 400 μM, PA 200 μM ± JTE-013 5 μM (S1PR2 antagonist), ± TY52156 5 μM (S1PR3 antagonist). In **H**, for each condition from left to right, *n* = 15, 5, 15, 15 and 15 separate images analyzed. **I** Average GP index from several images of NT, AdipoR2, S1PR2, S1PR3 siRNA HEK293 cells treated with PA 200 μM ± S1P 1 μM. For each condition from left to right, *n* = 10, 10, 15, 10, 14 and 15 separate images analyzed. Data are represented as mean ± SEM, except F: ± SD. *t*-tests (two sided and assuming normality) were used to identify significant differences between treatments. *$p < 0.05$, **$p < 0.01$, ***$p < 0.001$. See also Supplementary Fig. 5 and Supplementary Data 5. Source data are provided as a Source Data file.

2, and no expression of S1PR4-5 (Fig. 4F). Then, we selected specific S1PR agonist/antagonist (Fig. 4E) and assessed their effects on membrane health using Laurdan dye. TY52156, a specific S1PR3 antagonist caused PA sensitivity and increased membrane packing in MEFs, as indicated by a higher GP index (Fig. 4G, H). In contrast, SEW2871 and W146, S1PR1 agonist and antagonist respectively, or JTE-01, a S1PR2 antagonist, did not modify the MEFs response to PA (Fig. 4H and Supplementary Fig. 5H, I). To further strengthen our observations concerning the S1PRs, we carried out similar experiments in HEK293 and U-2 OS cells. Expression of S1PR1,2,3 and 5, but not S1PR4, was detected in HEK293 cells (Supplementary Fig. 5J). Again, the S1PR3 antagonist TY52156 enhanced the sensitivity of HEK293 cells to PA (Supplementary Fig. 5K), while the S1PR2 antagonist JTE-013 and the S1PR5 agonist A971432 did not influence membrane packing (Supplementary Fig. 5K, L). Equivalent experiments with the U2 O-S cells produced results similar to those in HEK293 cells. Specifically, we detected the expression of S1PR1, 2, 3 and 5 in U-2 OS cells (Supplementary Fig. 5M) and TY52156 was the only S1PR pharmacological intervention able to modify membrane packing (Supplementary

Fig. 5N). Next, we took a genetic approach and used siRNA against S1PR1-3 in HEK293 (Supplementary Fig. 5O–Q). In agreement with the pharmacological treatments, silencing of S1PR3 increased membrane packing (Fig. 4I) while S1PR1 or S1PR2 siRNA did not (Fig. 4I and Supplementary Fig. 5R).

S1PRs are G-protein-coupled receptors, and S1PR3 is able to couple with three different G-protein classes and thus activate multiple downstream effectors. More specifically, S1PR3 activation leads to the phosphorylation of PI3K and AKT[18], with AKT being a well-known activator of the sterol regulatory element-binding protein-1 (SREBP1)[39]. SREBP1 is a transcription factor that controls the expression of enzymes required for endogenous cholesterol, fatty acids, and phospholipid synthesis[40]. We have previously shown that SREBF1 silencing reduces *SCD* expression, which results in membranes rich in SFA-containing phospholipids and membrane rigidification[2]. Here, we first treated WT MEFs with betulin and confirmed that SREBP inactivation increases membrane packing in these cells (Fig. 5A). The SREBP pathway is evolutionarily conserved and, in *C. elegans*, a partial loss of function of SBP-1 (SBP-1 is the SREBP1 ortholog in *C. elegans* and null

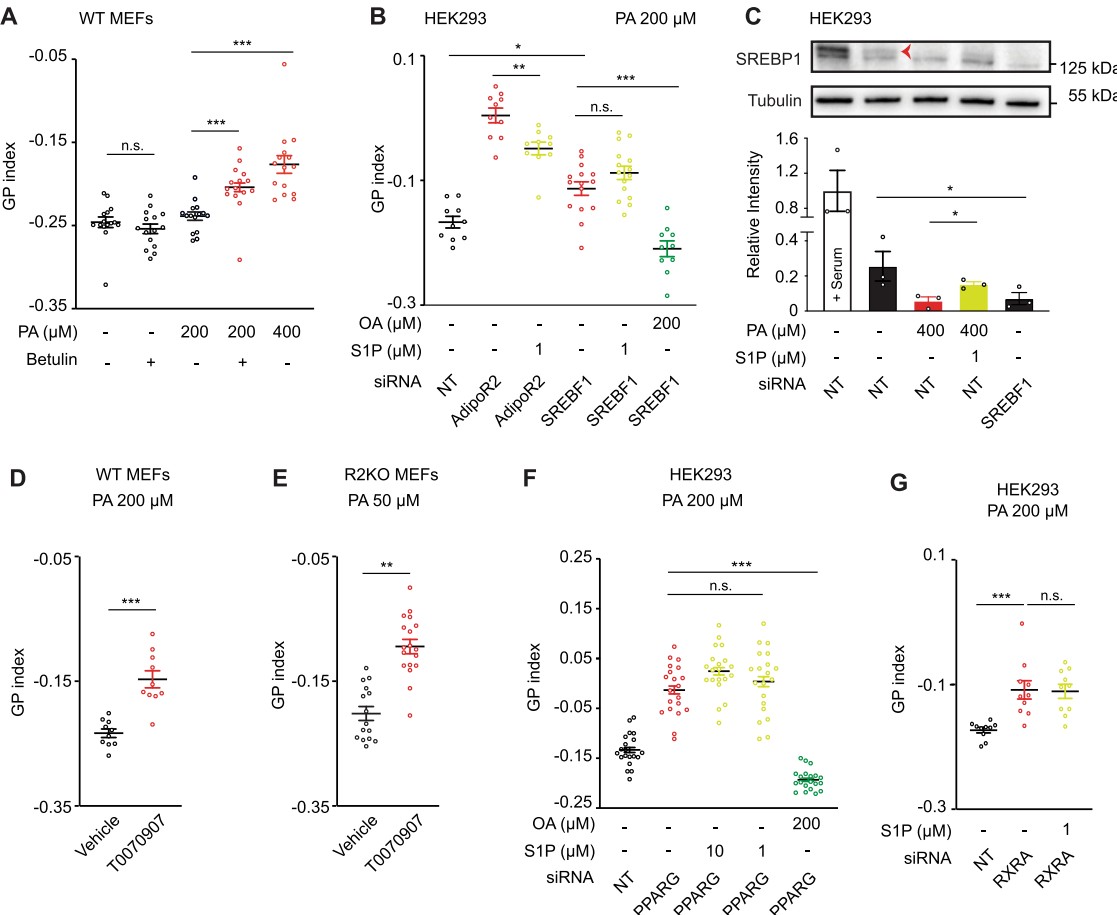

**Fig. 5 | S1P signaling via SREBP1 and PPARγ maintains membrane homeostasis in human/mouse cells. A** Average GP index from several images of WT MEFs treated with vehicle, betulin 1 μM (inhibitor of SREBP maturation) and PA 200 μM ± betulin 1 μM. *n* = 15 separate images analyzed for each condition. **B** Average GP index from several images of NT, AdipoR2 and SREBF1 siRNA HEK293 cells treated with PA 200 μM ± S1P 1 μM and ± OA 200 μM. For each condition from left to right, *n* = 10, 10, 10, 15, 15 and 10 separate images analyzed. **C** Western-Blot and quantification (*n* = 3 experiments) of NT and SREBF1 siRNA HEK293 cells treated with vehicle and PA 400 μM ± S1P 1 μM. The red arrow points to the precursor SREBP1 band. Note the presence of an unspecific band just below SREBP1 band. *n* = 3 biologically independent replicates per condition. Uncropped blots in Source Data. **D, E** Average GP index from several images of WT and R2KO MEFs

treated with vehicle, PA (200 or 50 μM) ± T0070907 1 μM (PPARγ antagonist). In **D**, *n* = 10 separate images analyzed. In **E**, *n* = 15 for vehicle and 18 for T0070907. Representative images of **D** are shown in Supplementary Fig. 7A. **F** Average GP index from several images of NT and PPARG siRNA HEK293 cells treated with vehicle and PA 200 μM ± S1P 10 or 1 μM. *n* = 20 separate images analyzed for each condition. **G** Average GP index from several images of NT and RXRA siRNA HEK293 cells treated with vehicle and PA 200 μM ± S1P 1 μM. Data are represented as mean ± SEM. *t*-tests (two sided and assuming normality) were used to identify significant differences between treatments. \**p* < 0.05, \*\**p* < 0.01, \*\*\**p* < 0.001. See also Supplementary Figs. 6, 7 and Supplementary Data 5. Source data are provided as a Source Data file.

alleles are lethal) caused PA sensitivity: *sbp-1* mutant worms grew slower and showed a tail tip defect similar to that of *paqr-2* mutants when grown on PA-loaded bacteria plates (Supplementary Fig. 6A, B and Supplementary Data 5). Next, we found that supplementation of S1P could not rescue the membrane packing defects caused by the loss of SREBP1 (Fig. 5B and Supplementary Fig. 6C), which is consistent with SREBP1 acting downstream of AdipoR2 and S1P. We then sought to determine whether S1P modulates SREBP1 processing. PA treatment reduced the amount of the unprocessed SREBP1 form (the inactive form of the transcription factor) in HEK293 cells, which was partially restored by S1P (Fig. 5C). Similarly, the levels of unprocessed SREBP1 were reduced in AdipoR2-lacking cells, but was boosted by S1P treatment (Supplementary Fig. 6D).

Beyond the similar membrane packing defects, the transcriptome of AdipoR1/2 and SREBP1/2 deficient cells should be similarly affected if both proteins are in the same pathway. To address this question, we compared the RNAseq profiles of HEK293 cells lacking AdipoR2[2] or SREBP1/2[41] (Supplementary Fig. 6E–J) in basal media and after PA treatment. 84 SREBP-regulated genes separated WT and AdipoR2-KO

cells after PA treatment in a PCA plot (Supplementary Fig. 6E). Importantly, the majority of those SREBP-regulated genes (77.4%) were also AdipoR2-regulated genes (Supplementary Fig. 6F), and the concordance was greatest among the most misregulated genes: i.e., SCD and INSIG (lipid synthesis) and HERPUD1 and JUN (ER-stress) (Supplementary Fig. 6G–J).

Altogether, these results indicate that AdipoR2-derived S1P acts via S1PR3 and SREBP1 to promote fatty acid desaturation and adequate membrane fluidity.

The transcription factor PPARγ is a master regulator of lipid metabolism[42] and is one of the intracellular targets of S1P[22,38]. More specifically, S1P binding to PPARγ promotes its binding to peroxisome-proliferator response elements and subsequent transcription[43]. Therefore, we next investigated the possible role of PPARγ in membrane fluidity regulation and its connection with AdipoR2 and S1P. MEFs, either WT or AdipoR2-KO, were more sensitive to low doses of PA when treated with the PPARγ antagonist T0070907 and showed increased GP index (Fig. 5D, E and Supplementary Fig. 7A). Likewise, the membranes of U-2 OS cells were excessively packed after

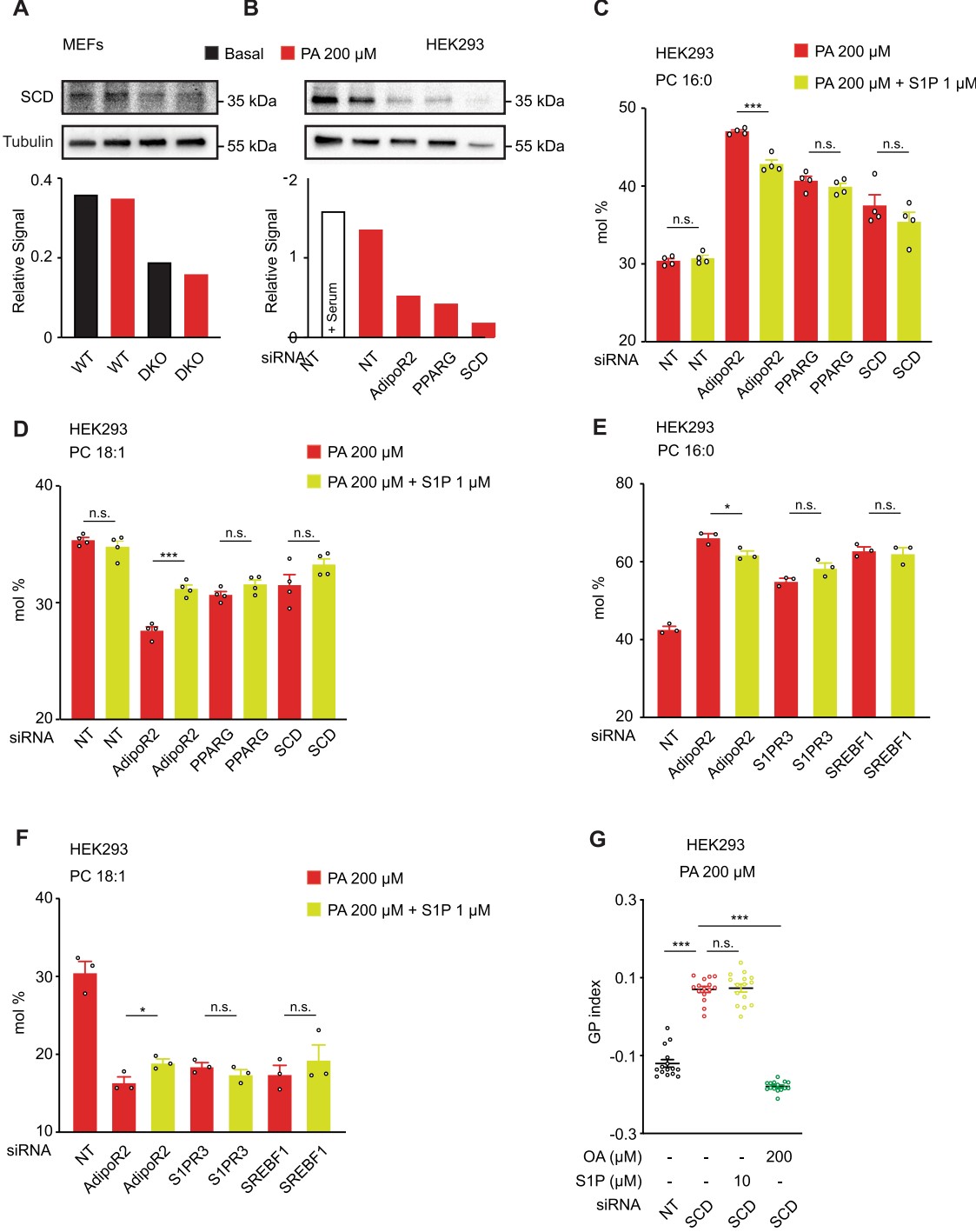

**Fig. 6 | SCD is the final protein effector of the AdipoR2-S1P pathway.**
**A, B** Western-Blot and quantification of WT and DKO MEFs in A and NT and Adi-poR2/PPARγ/SCD siRNA in HEK293 cells in full media or basal media ± PA 200 μM. Uncropped blots in Source Data. **C–F** PA (16:0) and OA (18:1) abundance (mol%) and in the PC of HEK293 cells treated with different siRNA and PA 200 μM ± S1P 1 μM. For C and D, $n = 4$ independent biological replicates per condition. For **E** and **F**, $n = 3$ independent biological replicates per condition. Related lipidomics are shown in

Supplementary Fig. 8M, N and in Data 6. **G** Average GP index from several images of NT and SCD siRNA HEK293 cells treated with vehicle and PA 200 μM ± S1P 10 μM and ± OA 200 μM. $n = 15$ separate images analyzed for each condition. Data are represented as mean ± SEM. $t$-tests (two sided and assuming normality) were used to identify significant differences between treatments. $*p < 0.05$, $**p < 0.01$, $***p < 0.001$. See also Supplementary Fig. 8 and Supplementary Data 5–7. Source data are provided as a Source Data file.

T0070907 treatment (Supplementary Fig. 7B). Next, we silenced the expression of PPARG in HEK293 cells (Supplementary Fig. S7C) and quantified membrane fluidity by FRAP and Laurdan. Both techniques showed that PPARγ-deficient cells had more rigid membranes when exposed to PA and that the addition of S1P to the culture medium was not sufficient to rescue this membrane phenotype (Fig. 5F and

Supplementary Fig. 7D). Importantly, PPARs function as obligate het-erodimers with other nuclear transcriptions factors such as retinoid X receptor (RXR)[42] and, as expected, the loss of RXRα mimicked the absence of PPARγ: we noticed excessive membrane packing upon silencing of RXRA and importantly, this could not be rescued by S1P (Fig. 5G, Supplementary Fig. 7E and Supplementary Data 5). Together,

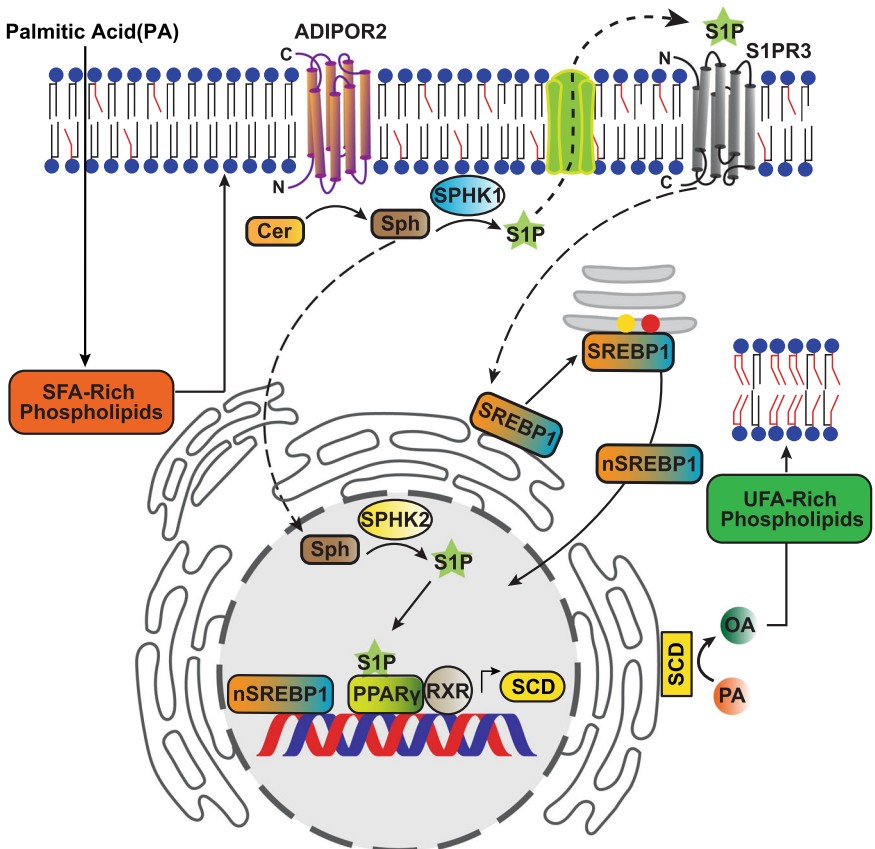

**Fig. 7 | Model.** Membrane rigidification activates a ceramidase activity intrinsic to AdipoR2, resulting in the generation of sphingosine 1-phosphate (S1P) that separately activates the SREBP1 and PPARγ transcription factors. SREBP1 and PPARγ together promote increased transcription of SCD, resulting in increased levels of unsaturated fatty acids and restored membrane fluidity.

these results are consistent with PPARγ/RXRα acting downstream of AdipoR2 and S1P.

Given that AdipoR2 has been shown to be associated with increased activity of another member of the PPAR family, namely PPARα[7], we explored whether there was any connection between PPARα and membrane fluidity regulation. For that, we used siRNA to silence PPARα in HEK293 cells (Supplementary Fig. 7F) followed by FRAP and Lipidomics. We found that the absence of PPARα did not alter membrane fluidity or phospholipid composition (Supplementary Fig. 7H–L), which indicates that PPARα is not part of the mechanisms responsible for membrane fluidity regulation.

In evolutionary terms, it has been shown that NHR-49 in *C. elegans* has a prominent role in modulating lipid metabolism and is typically referred to as a functional ortholog of PPARα in that organism[44,45], although NHR-49 is still considered to be an orphan receptor. Worms lacking NHR-49 recapitulated several *paqr-2* mutant phenotypes: they had rigid membranes, grew slower on PA-loaded bacteria and had the characteristic tail tip morphology defect (Supplementary Fig. 7M–O). To test whether S1P may function as an NHR-49 ligand, we first tagged the endogenous NHR-49 with GFP by using CRISPR/Cas9 genome editing. Fluorescent microscopy revealed the nuclear localization of this NHR-49 reporter in the intestinal and hypodermal cells, further confirmed by colocalization with the nuclear marker tdTomato::H2B (Supplementary Fig. 7P–S). Interestingly, worms grown on normal plates presented a small percentage of cells with a distinct focus within the nucleus (Supplementary Fig. 7R), which is likely the nucleolus. We then found that WT worms grown on PA-loaded bacteria responded by increasing the number of nuclei with such foci, but that *paqr-2* mutant was unable to replicate this dynamic translocation (Supplementary Fig. 7T). Importantly, the addition of S1P to the plate boosted the percentage of nuclei with foci in *paqr-2* mutants both at room temperature and at 15 °C (Supplementary Fig. 7U, V). Altogether, our experiments indicate that NHR-49 is the functional ortholog of PPARγ with respect to membrane homeostasis and that its sub-nuclear localization can be modulated by S1P.

In summary, we conclude that AdipoR2-derived S1P signals in parallel through S1PR3-SREBP and PPARγ to promote membrane homeostasis and that this pathway is evolutionary conserved.

## SCD-dependent desaturation as final output of AdipoR2-S1P signaling

The Δ−9 desaturase SCD is responsible for converting SFA into MUFA and is essential to sustain the levels of unsaturated fatty acids required for membrane homeostasis[2,24]. Remarkably, an RNAseq experiment previously showed that SCD is the most robustly downregulated gene in AdipoR2-KO HEK293 cells[2]. Here, we confirmed that DKO MEFs and HEK293 cells lacking AdipoR2 or PPARγ have reduced SCD at the protein and mRNA levels (Fig. 6A, B and Supplementary Fig. 8A, B). Likewise, in *C. elegans*, it has been shown that *paqr-2* mutant worms have reduced expression of the SCD ortholog *fat-7*[11]. Here, we confirmed that *fat-7* is downregulated in *paqr-2* mutant worms (Supplementary Fig. 8C–E) and also quantified the expression of the other two Δ−9 desaturases in *C. elegans*, namely *fat-5* and *fat-6*, and found that these were also downregulated in the *paqr-2* mutant (Supplementary Fig. 8F, G). Consistently with our model, other mutations/siRNA in the AdipoR2-S1P pathway caused down-regulation of Δ−9 desaturases. More specifically, *sphk-1, sbp-1, nhr-49* mutations in worms reduced the expression of Δ−9 desaturases (Supplementary Fig. 8D–G). As a consequence of the reduced desaturase expression, the *paqr-2, sbp-1* and *nhr-49* mutants all had increased amounts of SFA in membrane

phospholipids at the expense of PUFA (Supplementary Fig. 8H, I and Supplementary Data 7) (note that *C. elegans* have the ability to synthesize PUFA and hence 18:2 and 18:3 are not essential fatty acids as in humans[46]). Importantly, S1P boosted *fat-7* expression in *paqr-2* mutants (Supplementary Fig. 8J–L), which likely contributes to its above-described ability to rescue several membrane related phenotypes (Supplementary Fig. 3S–X).

To further verify the role of SCD as final output of the AdipoR2-S1P pathway, we assessed the importance of SCD activity using lipidomics analysis. In agreement with our model, the loss of AdipoR2 (upstream of S1P) resulted in an excess of SFA that was partially corrected by S1P, which increased the total amount of MUFA in PE and OA in PC at the expense of PA. However, the absence of S1PR3, SREBP1, PPARγ or SCD (downstream of S1P) resulted in an excess of SFA that was not rescued by S1P (Fig. 6C–F, Supplementary Fig. 8M, N and Supplementary Data 6). Accordingly, the Laurdan dye staining and FRAP assay showed that S1P did not suppress the excessive lipid-packing/membrane rigidity caused by the lack of SCD (Fig. 6G and Supplementary Fig. 8O), indicating that SCD is necessary for the effects of S1P on membrane homeostasis.

## Discussion

Since the initial cloning of the AdipoR1/2[47] the field has been mostly focused on the antidiabetic effects of these proteins[6,7,48]. However, there is a mounting number of evidences showing that the primary and evolutionary conserved function of the AdipoR1/2 proteins is to regulate cell membrane composition to maintain its fluidity (reviewed in ref.[12]).

A major finding of our work is the elucidation of the detailed molecular pathway by which cells regulate membrane homeostasis. Our results are consistent with the previous hypothesis that S1P is a mediator of AdipoR1/2 function[14] and link the documented ceramidase activity of the AdipoR1/2[13] with its primary function as fluidity regulator[2,24]. In particular, we propose that AdipoR1/2-derived S1P leads to the activation of the transcription factors SREBP1 and PPARγ to upregulate SCD expression and hence to the promotion of membrane fluidity (Fig. 7). PPARγ is a well-established regulator of lipid and glucose metabolism, however, most studies mainly focused on the role of PPARγ as a transcriptional controller of adipocyte differentiation and immune response[42]. Here, we show that PPARγ controls membrane homeostasis in multiple cell types. This is particularly interesting given that the functions of the broadly expressed isoform of PPARγ are not fully understood. Our work now suggests that PPARγ can regulate lipid metabolism through two different processes: (1) increase fat storage by defining adipocyte identity and (2) maintenance of membrane properties. Probably, these two outputs of PPARγ activation rely on different upstream activators and our work indicates that S1P is responsible for PPARγ-dependent promotion of membrane homeostasis.

DKO embryonic lethality in mice was previously reported[9], but not further investigated. We have systematically characterized DKO embryos using histology, proteomics and lipidomics. Our observations show that, in line with the earlier reports in cultured cells and *C. elegans*, the saturation of membrane phospholipids likely explains the DKO embryonic lethality. This highlights the vital importance of AdipoR1/2-mediated membrane homeostasis in a mammalian physiological context. Additionally, we also identified numerous metabolism-related defects in the DKO embryos, including reduced levels of apolipoproteins suggestive of poor lipid exchange among DKO tissues. Limited availability of ApoM, an S1P carrier in plasma, likely hampers S1P transport in the embryo. ApoM expression in mouse embryos is strongly induced at E11[49], suggesting that S1P transport is important from this stage. Membrane homeostasis can be regulated cell nonautonomously[25], and S1P could be acting as a signal molecule between tissues to globally maintain membrane homeostasis. Even

though there is a pronounced lipid imbalance in DKO embryos (Fig. 1D, E, and I, J and Supplementary Fig. 1I, J and L), a limitation of our study is that more mechanistic data are required to fully explain their regression/re-absorbance. However, it is noteworthy that ApoB ablation also causes embryonic lethality[50], as is also the case for PPARγ[51], SREBPs[52] and Sphk1/2 double null embryos[17]. The lethality of all these mutants demonstrates that membrane homeostasis is essential for the normal development of mouse embryos.

The present work leaves several questions open and has the limitation that it relied heavily on in vitro experiments. In the future it would be interesting to address the question of systemic membrane homeostasis with inducible KO mouse models or tissue-specific AdipoR-KOs to identify critical tissues. Additionally, the mechanism of membrane fluidity sensing by AdipoR2 was not addressed in the present work. The ER-resident protein RNF145, a ubiquitin ligase, promotes AdipoR2 degradation only under high membrane fluidity conditions, and this likely contributes to its regulation[53]. It would be interesting in the future to examine S1P signaling in animals where the expression of RNF145 is experimentally manipulated.

The AdipoR1-KO and AdipoR2-KO mice used in the present work were generated by Deltagen and have been studied in multiple laboratories[14,48,54–56]. However, these are not the only AdipoR knockouts mice described in the literature: another set of independently produced genomic lesions in the AdipoR1/2 genes has been generated and these DKO mice are viable[7]. Given that AdipoR proteins are undetectable in both null-mutant models[8,9,48] and considering the large number of genes participating in lipid metabolism, the different phenotypes could be explained by variations in the genetic background. Therefore, it would be interesting to test whether the other independently generated DKO mouse model also displays increased SFA content in membrane phospholipids accompanied by rigid membranes, though insufficient to cause lethality.

Although AdipoR1-KO embryos/MEFs showed a relatively mild membrane defect phenotype, the consequences of AdipoR1 mutations can still be dramatic in the adult or in specific tissues. For instance, a single amino acid mutation in AdipoR1 occurs in different forms of retinitis pigmentosa[57,58] and AdipoR1-KO mice have a substantial degeneration of photoreceptors and visual impairment as early as 3 weeks of age that is due to the lack of PUFA in membrane phospholipids[54,55]. This provides an independent confirmation that the AdipoR pathway is essential for membrane homeostasis, and shows that AdipoR1 has a specialized role in the maintenance of PUFA levels in the retina.

Finally, it will be of interest to understand whether our findings have therapeutic implications. For instance, pharmacological activation of AdipoR1/2 as nicely illustrated in[8,15] could be exploited to treat pathologies that correlate with abnormally elevated levels of SFA in phospholipids and membrane properties changes, such as diabetes (reviewed in[59]). However, the ability of the current AdipoR1/2 agonists to modulate membrane homeostasis remains to be investigated.

## Methods

### Mouse strains and husbandry

Heterozygous AdipoR1 and AdipoR2 gene knockout mice were obtained from Deltagen. Genomic lesions are described in[48]. Mutations were maintained on C57Bl/6 background. Experiments were approved by the Gothenburg Animal Ethics Committee (#2311/19). Mice were kept under temperature-controlled conditions with free access to water and standard rodent chow (Teklad 2016; Catalog: T.2016MI.12–16.4% of calories from protein and 4% from fat) with a 12 h light–12 h dark cycle. Mice were housed in Green Line individually ventilated cages at max 10 mice per cage. For breeding, two females and one male were placed in a cage. The pups were separated from the parents at 3 weeks of age. Mice had ad libitum access to food and water. Environmental enrichments such as wood

wool, nesting pad, chowing stick and cardboard boxes were provided in the cages.

## Experimental mouse crosses and embryo obtention
AdipoR1 and AdipoR2 heterozygous knockout animals were crossed to generate double heterozygotes that were further crossed to obtain E12.5 and E15.5 embryos. Tail biopsies were used for genotyping (see primers in list). Platinum™ II Hot-Start Green PCR Master Mix (2X) (Thermo Scientific) was used for the genotyping.

## Histology
E12.5 embryos were fixed in 4% buffered paraformaldehyde. Tissues were embedded in paraffin, sectioned, and stained with hematoxylin and eosin by Histocenter (Gothenburg, Sweden). Sections were scanned with a Zeiss Axio Scan.Z1 Slide Scanner.

## Cell culture conditions
Mouse Embryonic Fibroblast (MEFs) and HEK293 (ATCC, catalog#CRL-1573) cells were maintained in the DMEM medium (Gibco) containing glucose 1 g/L, pyruvate and GlutaMax and supplemented with 10% fetal bovine serum (HyClone) and 1% non-essential amino acids, HEPES 10 mM and 1% penicillin and streptomycin (all from Life Technologies). Human U-2 OS cells were cultured similarly to HEK293 except that glucose was at 4.5 g/L. Jurkat E6-1 were grown in RPMI 1640 medium with GlutaMax (Gibco) and supplemented with 10% fetal bovine serum and 1% penicillin and streptomycin. INS-1E cells were cultivated similarly to Jurkat with the addition of HEPES 10 mM and 0.05 mM 2-mercaptoethanol (Sigma). All cells were maintained at 37 °C in a water-humidified 5% $CO_2$ incubator. Cells were sub-cultured two or three times per week at 90% confluence, and cultivated on treated plastic flasks and multi-dish plates (Nunc). For FRAP and Laurdan dye experiments, cells were seeded in glass bottom dishes (Ibidi).

## MEFs isolation
The body of E12.5 embryos was maintained in cold PBS during genotyping and then tissues of interest were digested with Trypsin (Sigma) for 5–10 min at 37 °C and homogenized by pipetting and seeded in a 6 well-plate for 24 h and further expanded.

## Gene silencing in HEK293 cells
The following pre-designed siRNAs were purchased from Dharmacon: AdipoR2, Non-Target, PPARA, PPARG, RXRA, S1PR1, S1PR2, S1PR3, SCD, Sphk1, Sphk2, SREBF1 and SREBF2 (catalog numbers in Supplementary Table 1—see supplementary information file). Transfection of 25 nM siRNA was performed in complete media using Viromer Blue according to the manufacturer's instructions 1X (Lipocalyx). Knockdown gene expression was verified by qPCR 48 h after transfection.

## Fatty acid and Sphingosine 1-Phosphate (S1P) treatments: cell cultures
Palmitic acid (PA) and oleic acid (OA) were dissolved in sterile DMSO then mixed with fatty acid-free bovine serum albumin (BSA) (all from Sigma) in serum-free medium for 15 min at room temperature[2,10]. The molecular ratio of BSA to PA was 1–5.3 when using 400 mM palmitic acid, and 1–2.65 when using 200 mM PA. Cells were then cultivated in this serum-free media containing PA for 6, 18 or 24 h prior to analysis. An equivalent percentage of DMSO was used as a vehicle control in the experiments. S1P (Avanti Lipids) was dissolved in serum-free medium containing fatty acid free BSA and sonicated in a water bath at 37 °C. S1P aliquots were prepared at 100 µM and stored at −80 °C.

## Chemical agonist/antagonist
A971432 (S1PR5 agonist), JTE013 (S1PR2 antagonist), SEW2871 (S1PR1 agonist), TY52156 (S1PR3 antagonist), T0070907 (PPARγ antagonist) (all from Tocris) and betulin (inhibitor of SREBP maturation, from Sigma) were dissolved in DMSO. W146 (S1PR1 antagonist) was dissolved in ethanol. An equivalent percentage of DMSO/ethanol was used as a vehicle control in the experiments.

## Pre-Loading of *E. coli* with PA
Stocks of 100 mM PA dissolved in ethanol were diluted in LB media to final concentrations of 2 mM, inoculated with OP50 bacteria, then shaken overnight at 37 °C. The bacteria were then washed twice with M9 to remove unloaded fatty acids and growth media, diluted to equal OD600, concentrated 10 times by centrifugation, resuspended in M9 and seeded onto NGM plates lacking peptone[24].

## Electron Microscopy (EM) of MEFs
MEFs were prepared using high-pressure freezing followed by freeze substitution. Cells were centrifuged into a sealed pipet tip for better concentration of sample before transfer into the carrier[60]. Samples were then loaded into aluminum specimen carriers (0.1 mm cavity) with a hexadecane-coated flat carrier as a lid and were high-pressure frozen in a Wohlwend Compact 3 machine. A short freeze substitution protocol was applied, using 2% uranyl acetate dissolved in acetone (UA; from 20% UA stock in methanol) for 1 h[29,61]. The UA incubation was followed by two washes in 100% acetone, first wash for 1 h and last wash over-night. Simultaneously, the temperature was raised from −90 °C to −50 °C overnight with a rate of 3 °C/h. Samples were then infiltrated with HM20 resin in increasing concentrations of 20%, 40%, 50%, 80% in acetone and finally three times in 100% resin (2 h per solution). Polymerization of the plastic occurred over 48 h using UV light at −50 °C followed by 48 h in room temperature. Samples were sectioned in 70 nm thin sections and placed on copper slot grids. Sections were stained with 2% UA for 5 min and Reynold's lead citrate for 1 min[62]. Grids were washed in 3 x $dH_2O$ between each staining solution. All thin sections were imaged at 120 kV on a Tecnai T12 transmission electron microscope equipped with a Ceta CMOS 16 M camera (Thermo Scientific). Images were analyzed and prepared using the IMOD software package[63]. The distance of the closest ER-mitochondria was measured in nm, considering only the ER that was at a maximum range of 300 nm. Distances <2 nm were considered contact sites.

## Lipidomics
**Sample preparations.** Mouse embryos were extracted and maintained on ice until the head/brain was collected and immediately frozen in dry ice. Tissues were stored at −80 °C until analysis. MEFs and HEK293 cells (prepared in at least three independent replicates) were cultivated in basal and the presence of PA for 18 or 24 h prior to harvesting using TrypLE Express (Gibco), washed with PBS and stored at −80 °C until analysis. For worms lipidomics, samples were composed of synchronized L4 larvae (one 9 cm diameter plate/ sample; each treatment/ genotype was prepared in four independently grown replicates) grown on non-peptone plates seeded with OP50 and OP50 pre-loaded with PA. Worms were washed three times with M9, pelleted and stored at −80 °C until analysis. For lipid extraction, cells and worms were sonicated for 10 min in methanol and then extracted according to published methods[64]. Tissues were homogenized in butanol/methanol [3:1 v/v] and extracted according to a published method[64]. Internal standards were added during the extraction.

**Lipid analysis.** Several different platforms were used for targeted lipid analysis. Ceramides, dihydroceramides, glucosylceramides and lactosylceramides were quantified using UPLC-MS/MS on a QTRAP 5000 (Sciex) in positive ESI mode similar to what has previously been described[65]. For this, a part of the lipid extract was reconstituted in chloroform:methanol:water [3:6:2; v/v/v] and 5 µl was injected into the UPLC-MS/MS system. Molecular species were separated on a BEH C8 column (2.1 × 100 mm with 1.7 µm particles) from Waters kept at 60

degrees. The mobile phases were water:acetonitrile [70:30; v/v] with 0.1% formic acid as A-phase and acetonitrile:isopropanol [1:1; v/v] with 0.1% formic acid as B-phase. The separation was done at 400 μl/min using a linear gradient from 75 to 100% B-phase over 5 min. The mobile phase composition was the kept at 100% B for 2 min and then returned to 75% B for 3 min for a total runtime of 10 min. Quantification of 27 lipid species was done using an external calibration curve made from 16 reference substances (six ceramides, four dihydroceramides, three glucosylceramides and three lactosylceramides). For accurate quantification internal standards (C17:0 ceramide, C17:0 glucoslyceramide and C17:0 lactosylceramide) were added during extraction. Instrument parameters were: CUR = 20, CAD = 7, IS = 4500, TEM = 350, GS1 = 60, GS2 = 60, DP = 100, EP = 10 and CXP = 26. Collision energies and MRM-transitions were optimized for each available reference substance.

Sphingosine (Sph) and sphingosine-1-phosphate (S1P) were analyzed using UPLC-MS/MS on a QTRAP 5000 (Sciex) in positive ESI mode similar to what has previously been described[65]. For this a part of the lipid extract was reconstituted in methanol:acetonitrile:water [2:1:1] with 0.1% formic acid and 5 μl was inject and separated using a Kinetex C8 column (2.1 × 100 mm with 1.7 μm particles) from Phenomenex kept at room temperature. The mobile phases were water:acetonitrile [40:60] with 5 mM ammonium formate and 0.1% formic acid as A-phase and acetonitrile with 5 mM ammonium formate and 0.1% formic acid as B-phase. The separation was performed at 300 μl/min using a linear gradient from 0 to 50% B-phase over 5 min. The mobile phase composition was then increased to 100% B and held for 2 min before returning to 40% B for 3 min for a total runtime of 10 min. Quantification was made using external standard curve made from reference substances. For accurate quantification, internal standards (D$_7$-Sph and $^{13}$C$_2$D$_2$-S1P) were added during extraction. Instrument parameters were: CUR = 20, CAD = 7, IS = 4500, TEM = 500, GS1 = 60, GS2 = 60, DP = 100, EP = 10 and CXP = 26. Collision energies were 20 volts for S1P and 13 volts for Sph. The MRM transition used for Sph and D$_7$-Sph were 300.4 > 282.4 and 307.4 > 289.4 respectively. For S1P and $^{13}$C$_2$D$_2$-S1P the transitions 380.4 > 264.4 and 384.4 > 268.4 were used.

Phospholipids, sphingomyelins and triglycerides were all analyzed using direct infusion (shotgun) mass spectrometry on a QTRAP 5500 mass spectrometer (Sciex)[66–68]. For this a part of the lipid extract was evaporated and reconstituted in chloroform:methanol [1:2; v/v] with 5 mM ammonium acetate. For sphingomyelin, the extract was exposed to alkaline hydrolysis (0.1 M KOH in methanol at room temperature for 60 min) prior to reconstitution in order to remove mass-interfering phospholipids. Infusion was made using a TriVersa Nano-Mate interface (Advion Bioscience) working in positive mode using 1.2 kV and 0.8 psi of nitrogen gas pressure. Precursors of phosphatidylcholine and sphingomyelin were scanned between 600-850 m/z and detected using the phosphocholine fragment at m/z 184. The phosphatidylethanolamines were detected using neutral loss of m/z 141. Precursors were scanned between 650-850 m/z. Triglycerides were detected using multiple neutral loss scanning[69]. Precursors were detected in the range of 800-1000 by monitoring the loss of several fatty acid fragments (between C14:0 to C22:6). The collision energies were optimized per lipid class and were (in volt): PC = 45, PE = 30 SM = 45, TG = 35. To achieve a robust signal with sufficient ion statistics, a minimum of 60 scanning cycles were performed for each lipid class. The analyses were performed at a scan rate of 200 Da/s with a stepsize of 0.1 Da.

The generated raw data files (.wiff files) were processed using the LipidView and MultiQuant softwares (Sciex). Identified lipids were quantified against the signal from the lipid-class specific internal standards, which were added during extraction. The internal standards used were d6-TG 16:0/16:0/16:0, PC 17:0/17:0, PE 17:0/17:0 and SM 12:0.

Qlucore Omics Explorer software was used for the multivariant analysis. The complete lipid composition data are provided in supplementary data files and uploaded to Zenodo with dataset identifier 10.5281/zenodo.7024516.

## Proteomics

**Sample preparation for proteomic analysis.** Samples were homogenized on a FastPrep-24 instrument (MP Biomedicals) for five repeated 40 s cycles at 6.5 m/s in lysis buffer containing 2% SDS, 50 mM triethylammonium bicarbonate (TEAB). Lysed samples were centrifuged at 21100 g for 10 min and the supernatants were transferred to clean tubes. Protein concentrations were determined using Pierce BCA Protein Assay Kit (Thermo Scientific) and a Benchmark Plus microplate reader (Bio-Rad). Aliquots containing 30 μg of total protein from each sample were incubated at 56 °C for 30 min in the lysis buffer with DTT at 100 mM final concentration. The reduced samples were processed using the modified filter-aided sample preparation (FASP) method[70]. In short, the reduced samples were diluted to 1:4 by 8 M urea solution, transferred onto Nanosep 30k Omega filters (Pall Corporation) and washed repeatedly with 8 M urea and once with digestion buffer (0.5% sodium deoxycholate in 50 mM TEAB). Free cysteine residues were modified using 10 mM methyl methanethiosulfonate (MMTS) solution in digestion buffer for 20 min at RT and the filters were washed twice with 100 μl of digestion buffer. Pierce trypsin protease (Thermo Scientific) in digestion buffer (0.5% sodium deoxycholate in 25 mM TEAB) was added at a ratio of 1:100 relative to total protein mass and the samples were incubated at 37 °C for 3 h. An additional portion of trypsin was added and incubated overnight. The peptides were collected by centrifugation and isobaric labeling was performed using Tandem Mass Tag (TMT-10plex) reagents (Thermo Scientific) according to the manufacturer's instructions. The labelled samples were combined into one pooled sample, concentrated using vacuum centrifugation, and SDC was removed by acidification with 10% TFA and subsequent centrifugation. The labelled pooled sample was treated with Pierce peptide desalting spin columns (Thermo Scientific) according to the manufacturer's instructions. The purified desalted sample was pre-fractionated into 40 primary fractions with basic reversed-phase chromatography (bRP-LC) using a Dionex Ultimate 3000 UPLC system (Thermo Scientific). Peptide separations were performed using a reversed-phase XBridge BEH C18 column (3.5 μm, 3.0 × 150 mm, Waters Corporation) and a linear gradient from 3% to 40% solvent B over 18 min followed by an increase to 100% B over 5 min and 100% B for 5 min at a flow of 400 μL/min. Solvent A was 10 mM ammonium formate buffer at pH 10.00 and solvent B was 90% acetonitrile, 10% 10 mM ammonium formate at pH 10. The fractions were concatenated into 20 fractions, dried and reconstituted in 3% acetonitrile, 0.2% formic acid.

**nLC-MS/MS.** The fractions were analyzed on an orbitrap Fusion Lumos Tribrid mass spectrometer interfaced with Easy-nLC1200 liquid chromatography system (Thermo Scientific). Peptides were trapped on an Acclaim Pepmap 100 C18 trap column (100 μm × 2 cm, particle size 5 μm, Thermo Scientific) and separated on an in-house packed analytical column (75 μm × 35 cm, particle size 3 μm, Reprosil-Pur C18, Dr. Maisch) using a gradient from 5% to 12% B over 5 min, 12–5% B over 72 min followed by an increase to 100% B for 3 min, and 100% B for 10 min at a flow of 300 nL/min. Solvent A was 0.2% formic acid and solvent B was 80% acetonitrile, 0.2% formic acid. MS scans were performed at 120,000 resolution, m/z range 375-1375. MS/MS analysis was performed in a data-dependent, with top speed cycle of 3 s for the most intense doubly or multiply charged precursor ions. Precursor ions were isolated in the quadrupole with a 0.7 m/z isolation window, with dynamic exclusion set to 10 ppm and duration of 45 s. Isolated precursor ions were subjected to collision induced dissociation at 35 collision energy with a maximum injection time of 50 ms. Produced MS2 fragment ions were detected in the ion trap followed by multi-notch (simultaneous) isolation of the top 10 most abundant fragment

ions for further fragmentation (MS3) by higher-energy collision dissociation (HCD) at 65% and detection in the Orbitrap at 50 000 resolutions, $m/z$ range 100–500.

**Proteomic data analysis.** The data files were merged for identification and relative quantification using Proteome Discoverer v.2.4 (Thermo Scientific). Swiss-Prot *Mus musculus* database was used for the database search, using the Mascot search engine v.2.5.1 (Matrix Science) with MS peptide tolerance of 5 ppm and fragment ion tolerance of 0.5 Da. Tryptic peptides were accepted with 0 missed cleavage and methionine oxidation was set as a variable modification. Cysteine methylthiolation and TMT on peptide N-termini and on lysine side chains were set as fixed modifications. Percolator was used for PSM validation with the strict FDR threshold of 1%. Quantification was performed in Proteome Discoverer 2.4. The TMT reporter ions were identified with 3 mmu mass tolerance in the MS3 HCD spectra and the TMT reporter S/N values for each sample were normalized within Proteome Discoverer 2.4 on the total peptide amount. Only the quantitative results for the unique peptide sequences with the minimum SPS match % of 65 and the average S/N above 10 were taken into account for the protein quantification. Peptides were filtered for high confidence. Qlucore Omics Explorer software was used for the multivariant analysis. The complete proteomics dataset is provided in a supplementary datafile and have been deposited to the ProteomeXchange Consortium via the PRIDE partner repository with the dataset identifier PXD029163.

**Laurdan dye measurement of membrane packing (confocal microscopy)**
Live cells were stained with Laurdan dye (6-dodecanoyl-2-dimethylaminonaphthalene) (Thermo Scientific) at 15 μM for 45 min. MEFs, HEK293, INS-1E and U-2 OS cells were analyzed by confocal microscopy[2,10]. Images were acquired with an LSM880 confocal microscope equipped with a live cell chamber (set at 37 °C and 5% $CO_2$) and ZEN software with a 40X water-immersion objective. Cells were excited with a 405 nm laser, and the emission was recorded between 410 and 461 nm (ordered phase) and between 470 and 530 nm (disordered phase). Pictures were acquired with 16-bit image depth and 1024 × 1024 resolution using a pixel dwell of ~1.02 μs. Images were analyzed using ImageJ version 1.47 software following published guidelines[71].

**Laurdan dye measurement of membrane packing (flow cytometry)**
Live Jurkat E6.1 cells were stained with Laurdan dye as detailed above and analyzed by flow cytometry using a Cytoflex S (Beckman Coulter). Cells were resuspended in HBSS without $Ca^{2+}$ and $Mg^{2+}$ (Gibco) at pH 7.4 supplemented with 10 mM HEPES. Cells were excited with the UV laser (405 nm) and the emission was read with the fluorescent channel 450/45 BP for the ordered phase and with the 525/40 BP for the disordered phase. Data were analyzed with the Kaluza software (Beckman Coulter) version 2.1.

**Other fluorescent microscopy in MEFs**
MEFs were treated with PA 200 μM for 18 h and then stained with BODIPY 500/510 C1, C12 (4,4-Difluoro- 5-Methyl-4-Bora-3a,4a-Diaza-s-Indacene-3-Dodecanoic Acid) (Invitrogen) at 2 mg/ml in PBS for 10 min or with Laurdan (see above) ± ER-Tracker™ Red (BODIPY™ TR Glibenclamide) (Thermo Scientific) at 0.5 μM. Z-stacks were acquired with an LSM880 confocal microscope equipped with a live cell chamber (set at 37 °C and 5% CO2) and ZEN software (Zeiss) with a 40X water objective. 2.5D image reconstructions containing spatial information were made using the temporal color code tool in ImageJ software (v2.1). Lipid droplets were stained using LipidSpot 610 (Biotium) at 1X following manufacturer´s instructions.

**Fluorescence recovery after photobleaching (FRAP) in cells**
For FRAP in mammalian cells, HEK293 and INS-1E cells were stained with BODIPY 500/510 C1, C12 at 2 mg/ml in PBS for 10 min at 37 °C[10,24]. FRAP images were acquired with an LSM880 confocal microscope equipped with a live cell chamber (set at 37 °C and 5% $CO_2$) and ZEN software (Zeiss) with a 40X water-immersion objective. Cells were excited with a 488 nm laser and the emission between 493 and 589 nm recorded. Images were acquired with 16 bits image depth and 256 × 256 resolution using a pixel dwell of ~1.34 μs. Ten pre-bleaching images were collected and then the region of interest was beached with 50% of laser power. The recovery of fluorescence was traced for 25 s. Fluorescence recovery and $T_{half}$ (50% intensity of the fluorescence recovered) were calculated as in ref. 24.

**Quantitative PCR (qPCR)**
Total cellular/worm RNA was isolated using RNeasy Plus Kit according to the manufacturer's instructions (Qiagen) and quantified using a NanoDrop spectrophotometer (ND-1000; Thermo Scientific). cDNA was obtained using a RevertAid H Minus First Strand cDNA Synthesis Kit (Thermo Scientific) with random hexamers. qPCR was performed with a CFX Connect thermal cycler (Bio-Rad) using HOT FIREpol Eva-Green qPCR Supermix (Solis Biodyne) and standard primers (sequences in Supplementary Table 2 – see supplementary information fie). Samples were measured as triplicates. The relative expression of each gene was calculated according to the delta-delta CT method[72]. *PPIA* expression was used as calibrator in human and mouse experiments and *tba-1* in *C. elegans*.

**Protein extraction and western-blot**
Cellular proteins were extracted using "Lola's" lysis buffer (1% Nonidet P-40, 0.1% SDS, 10% glycerol, 1% sodium deoxycholate, 1 mM DTT, 1 mM EDTA, 100 mM HEPES, 100 mM KCl) containing Halt Protease Inhibitor Cocktail (1X; Pierce) on ice for 10 min. Upon lysis completion, cell lysates were centrifuged at 20000 g for 10 min at 4 °C. The soluble fraction was kept for further analysis, and the protein sample concentration was quantified using the BCA protein assay kit (Pierce) according to the manufacturer's instructions. Twenty (HA blot) or forty (SREBP1 blot) micrograms of protein were mixed with Laemmli sample loading buffer (Bio-Rad), heated to 37 °C for 10 min, and loaded in 4% to 20% gradient precast SDS gels (Bio-Rad). After electrophoresis, the proteins were transferred to nitrocellulose membranes using Trans-Blot Turbo Transfer Packs and a Trans-Blot Turbo apparatus/predefined mixed-MW program (Bio-Rad). Blots were blocked with 5% nonfat dry milk (HA blot) or EveryBlot Blocking Buffer (Bio-Rad) (SREBP1 blot) in PBS-T for 1 h at room temperature. Blots were incubated with primary antibodies overnight at 4 °C: rabbit monoclonal anti-HA antibody (C29F4, Cell signaling, catalog#14031) 1:5000 dilution, rabbit monoclonal anti-SCD (C12H5, Cell Signaling, catalog#2794) 1:1000 dilution, mouse monoclonal anti-SREBP1 (2A4, Santa Cruz, catalog#sc-13551) 1:200 or mouse monoclonal anti-Tubulin (B512, Sigma, catalog#5168) 1:5000. Blots were then washed with PBS-T and incubated with either swine anti-rabbit HRP (1:3000, Dako, catalog#P0399) or goat anti-mouse HRP (1:3000, Dako, catalog#P0447) and washed again with PBS-T. Detection of the hybridized antibody was performed using an ECL detection kit (Immobilon Western, Millipore), and the signal was visualized with a digital camera (ChemiDoc XRS + , Bio-Rad). Signal intensities were quantified using Image Lab Software (Bio-Rad). Uncropped blots are shown in source data files and supplementary information.

**Glucose Stimulated Insulin Secretion (GSIS)**
INS-1E cells were cultivated overnight in serum-free medium ± PA 400 μM ± S1P, washed with Hanks' Balanced Salt Solution (HBSS) without $Ca^{+2}$ and $Mg^{+2}$ and maintained in HBSS without glucose for 2 h. Then, the cells were stimulated with HBSS with glucose 16 mM for

30 min and the supernatant later collected. The amount of insulin secreted was quantified using a Rat Insulin ELISA (Mercodia) following manufacturer's instructions.

## Plasmids for HEK293 cells and transfection

pIRESHyg2-HA-hAdipoR2-cMYC construct was described in[36]. pIRE-Shyg2-HA-hAdipoR2-cMYC(H348A) construct was generated using PCR-based mutagenesis (Q5-site-directed mutagenesis kit, New England Biolabs) with the following primers: 5'-gctcagctgtttcatatctttg-3' and 5'-agagtgaaaccagatgtcac-3'. pIRESHyg2-HA-hAdipoR2-cMYC(H202A, H348A, H352A) construct was generated using Gibson-assembly cloning kit (NEB) with the following primers: 5'-ttttcatggctcttcgccacagt ctactgccactcagagg-3' and 5'-agcaaacagctgagcagagtgaaaccagatgtcacatt tg-3' for amplification of the fragment generating H348A and H352A mutations; 5'-tctgctcagctgtttgctatctttgtggttgctggagc-3' and 5'-ggcgaag agccatgaaaaagaaaggcagagaatggctcct-3' for amplification of the fragment generating H202A mutation and the rest of the vector sequence. HEK293 cells were transfected using Viromer Red according to the manufacturer's instructions 1X protocol (Lipocalyx) and the protein expression verified by Western-blot 24 h after transfection.

## C. elegans strains

C. elegans strains were cultured as in[73]. The wild-type C. elegans reference strain N2, paqr-2(tm3410), sphk-1(ok1097), nhr-49(gk405) and sbp-1(ep79) and the transgene carrying strain EG7865 {oxTi617 [eft-3p::tdTomato::H2B::unc-54 3'UTR + Cbr-unc-119(+)]} are available from the Caenorhabditis Genetics Center (CGC; USA). The pfat-7::GFP(rtIs30) carrying strain HA1842 was a kind gift from Amy Walker[74].

The PHX3258 (nhr-49::GFP) strain was created by Suny Biotech (Fuzhou City, China) using CRISPR/Cas9 and carries a modified nhr-49 locus where the end of the coding region is fused in-frame with that of GFP. The altered sequence is as follows (underlined sequences are from the endogenous nhr-49, linker sequences are in bold, GFP coding sequences are in regular uppercase, introns and 3'UTR are in lowercase, and the STOP codon is in italics): <u>TTGAACAGTGAGCAGAAT AATCATATGCTC</u>**AGTAAAGGAGAAGAACT**TTTTCACTGGAGTTGTCC CAATTCTTGTTGAATTAGATGGTGATGTTAATGGGCACAAATTTTCTG TCAGTGGAGAGGGTGAAGGTGATGCAACATACGGAAAACTTACCCTT AAATTTATTTGCACTACTGGAAAACTACCTGTTCCATGGgtaagtttaaaca tatatatactaactaaccctgattatttaaattttcagCCAACACTTGTCACTACTTT CTgTTATGGTGTTCAATGCTTcTCgAGATACCCAGATCATATGAAAC gGCATGACTTTTTCAAGAGTGCCATGCCCGAAGGTTATGTACAGGAA AGAACTATATTTTTCAAAGATGACGGGAACTACAAGACACgtaagtttaa acagttcggtactaactaaccatacatatttaaattttcagGTGCTGAAGTCAAGTTT GAAGGTGATACCCTTGTTAATAGAATCGAGTTAAAAGGTATTGATT TTAAAGAAGATGGAAACATTCTTGGACACAAATTGGAATACAACTATA ACTCACACAATGTATACATCATGGCAGACAAACAAAAGAATGGAAT CAAAGTTgtaagtttaaacatgattttactaactaactaatctgatttaaattttcagAACT TCAAAATTAGACACAACATTGAAGATGGAAGCGTTCAACTAGCAGAC CATTATCAACAAAATACTCCAATTGGCGATGGCCCTGTCCTTTTACC AGACAACCATTACCTGTCCACACAATCTGCCCTTTCGAAAGATCCCAA CGAAAAGAGAGACCACATGGTCCTTCTTGAGTTTGTAACAGCTGCTG GGATTACACATGGCATGGATGAACTATACAAA*TAA*taattccattttttctccc aaaactcttcacctcat. The PHX3258 strain is readily available from the authors and will be submitted to the Caenorhabditis Genetics Center (CGC).

## C. elegans culture conditions

Unless otherwise stated, experiments were performed at 20 °C, using the E. coli strain OP50 as food source, which was maintained on LB plates kept at 4 °C (re-streaked every 6–8 weeks), and single colonies were picked for overnight cultivation at 37 °C in LB medium before being used to seed NGM (nematode growth media) plates[75]. Plates containing glucose or S1P were prepared by adding the respective stock solutions to cooled NGM after autoclaving.

## FRAP in C. elegans

FRAP experiments in C. elegans were carried out using a membrane-associated prenylated GFP reporter expressed in intestinal cells and using a Zeiss LSM700 inv laser scanning confocal microscope with a 40X water-immersion objective. Briefly, the GFP-positive membranes were photobleached over a rectangular area (15 × 4 pixels) using 30 iterations of the 488 nm laser with 50% laser power transmission. Images were collected at a 12-bit intensity resolution over 256 × 256 pixels (digital zoom 4X) using a pixel dwell time of 1.58 µs, and were all acquired under identical settings. The recovery of fluorescence was traced for 25 s. Fluorescence recovery and $T_{half}$ were calculated as in ref. 76.

## C. elegans growth

For length measurement studies, synchronized L1s were plated onto test plates seeded with E. coli, and worms were mounted and photographed 144 h (15 °C experiments) or 72 h (all other experiments) later. The length of 20 worms were measured using ImageJ 2.1[77]. Alternatively, the stages of the worms (L1, L2-3, L4 or adult, n = 50).

## S1P plates and S1P injections in C. elegans

S1P (Avanti Lipids) was dissolved in butanol:methanol (3:1) and sonicated in a water bath at 55 °C. S1P stock solution (1.32 mM) was further stored at −80 °C. Plates containing S1P were prepared by adding stock solution of S1P to cooled NGM after autoclaving[78]. For injections, 1 µM S1P was directly injected into the C. elegans gonad.

## Quantification of tail tip morphology in C. elegans

Quantification of the withered tail tip phenotype was done on synchronous 1-day-old adult populations, that is 72 h post L1 (n ≥ 50)[11].

## C. elegans FAT-7 expression

Quantification of fluorescence intensity of pfat-7::GFP carrying strains were performed on L4 larvae using ImageJ 2.1 (n ≥ 20)[11].

## Statistics and reproducibility

Unless otherwise stated, each experiment presented here was reproduced independently at least three times with similar results, including the representative images shown in Figs. 2E–J; 3B; 4B, G; 6A, B and Supplementary Figs. 1a–h; 2f–k; 3a–c, p, r, s; 7a, p–s; 8c, j. Error bars show the standard error of the mean, and t-tests (two sided and assuming normality) were used to identify significant differences between treatments (except in qPCR panels, where error bars show standard error). For the lipidomics and proteomics analysis, ANOVA was used to identify lipids that were significantly different among the groups (including and adjustment for multiple comparisons q-value). For electron microscopy experiments, an exact binomial test was performed with a hypothesized probability of success equal to the frequency of the respective control group. The test was performed using the R package (R Core Team, 2018). In the box plots, boxes (with mean) indicate the 25th to 75th percentile while the whiskers indicate the data points still within 1.5 of the box range. Asterisks are used in the figures to indicate various degrees of significance, where *p < 0.05; **p < 0.01; and ***p < 0.001.

## Reporting summary

Further information on research design is available in the Nature Portfolio Reporting Summary linked to this article.

## Data availability

Source data generated in this study are provided in supplementary files/source data files and deposited in Zenodo with dataset identifier https://doi.org/10.5281/zenodo.7024516 [https://zenodo.org/record/ 7024516#.Y2q8sC8w1TZ] and in ProteomeXchange Consortium via

the PRIDE partner repository with the dataset identifier PXD029163. Source data are provided with this paper.

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

## Acknowledgements

We thank Rafael Camacho and the Centre for Cellular Imaging (CCI) at the University of Gothenburg and the National Microscopy Infrastructure, NMI (VR-RFI 2016–00968) for assistance in microscopy. The Proteomics Core Facility at Sahlgrenska Academy, University of Gothenburg performed the analysis for protein quantification. Some *C. elegans* strains were provided by the *Caenorhabditis* Genetics Center, which is funded by NIH Office of Research Infrastructure Programs (P40 OD010440). We also thank Clotilde Wiel and Volkan Sayin for helping with scanning the histological sections. Cristina Fernandez and Irene Cozar for the GSIS protocol. Beatriz Martinez-Abad and Malin Johansson for helping with the flow cytometry experiments. Anton Rydberg for technical help with some qPCRs and Valentina Sukonina, Sven Enerbäck, Lisa Nilsson, Jonas Nilsson, Andrej Besse and Lenka Besse for sharing different cell lines. This work was supported by: Swedish Research Council (Vetenskapsrådet Dnr: 2020-03300 and 2015-00560 M.P.), Cancerfonden (Dnr: 19 0029 Pj M.P.), Kungliga Vetenskaps- och Vitterhets-Samhället i Göteborg (2020–462 M.R.), Stiftelserna Wilhelm och Martina Lundgrens (2020–3610 M.R.) and Åke Wibergs Stiftelse (M20-0151 M.R.).

## Author contributions

Conceptualization, M.R., R.D., M.P.; investigation M.R., R.D., D.P., P.-O.B., D.K., M.H., A.N.; writing—original draft MR; writing—review and editing, M.R., M.P.; funding acquisition, M.R., J.B., M.P.; resources, K.P., J.L.H.,

M.B.-Y., P.C., J.B. All authors approved the final version of the manuscript.

## Funding

## Competing interests
K.P. and M.B. are presently employed by AstraZeneca and may be AstraZeneca shareholders. The remaining others declare no competing interests.
