## [Peer Review File · Nature Communications]

SPHINGOSINE 1-PHOSPHATE MEDIATES ADIPONECTIN RECEPTOR SIGNALING ESSENTIAL FOR LIPID HOMEOSTASIS AND EMBRYOGENESISREVIEWER COMMENTS

Reviewer #1 (Remarks to the Author):

Ruiz, Pilon and colleagues present a very dense data set with a lot of exciting data that will prove to be quite stimulating for the field. Based on data obtained from AdipoR1/2 double knock out embryos as well as cell lines and *C. elegans* studies, the authors show that double KO's display excessive membrane phospholipid saturation. This leads them to suggest that consistent with the previously identified enzymatic activity of the receptors, the AdipoR1/2 ceramidase activity converts ceramides (generated from exogenous palmitate) to sphingosine, which in turn gets converted to sphingosine phosphate, activating the S1P receptors. They show that this triggers activation of SREBP1, that activates together with S1P-activated PPARgamma the transcription of SCD1, which in turn mediates the desaturation of membrane phospholipids.

A very interesting set of observations that lead the authors to some interesting conclusions. While the proposed model may be somewhat oversimplified, most of the conclusions seem warranted base on the data that the authors present.

Given the data density, the paper is rather hard to follow. More importantly, the figures a extremely complex. While the rationale that the authors used to further underline their finding with studies in *C. elegans*, it may be better to remove the *C. elegans* data into a separate future paper and focus on the results in mammalian cells.

Despite the complexity of the data presented, there is a simple take home message for the reader, namely that the main cellular role of the adiponectin receptors is fatty acid desaturation of membrane phospholipids. That is an excellent postulate, and a lot of existing data in the field on adiponectin and its receptors may be explained on the basis of this mechanism.

A few things should be clarified specifically:

Fig.1D needs to be better explained. "Lipidomics analysis of the membrane phospholipids phosphatidylcholine (PC) and phosphatidylethanolamine (PE), triacylglycerols (TAG), free cholesterol (FC) and the sphingolipids sphingomyelin (SM), ceramide (Cer), dihydroceramide (DiCer), glucosylceramide (GluCer) and lactosylceramide (LacCer) allowed for a clear separation of WT, AdipoR1-KO, AdipoR2-KO and DKO E12.5 embryos in a principal component analysis

(PCA)". There has to be a better way to present that data!

It is interesting that exogenous S1P can reconstitute the AdipoR deficiencies with respect to membrane fluidity. In many instances, addition of exogenous S1P can drive substrate-driven reactions the other way. Does addition of S1P lead to increased ceramides? This should be determined in one of the systems used.

Perhaps the weakest link in their scheme relates to SREBP activation. Again, the C elegans data is not that helpful, and the authors should provide additional data in the mammalian system demonstrating the nature of the downstream SREBP activation.

Nevertheless, these points should not distract from the fact that this is an outstanding set of observations that significantly advance the field!

Reviewer #2 (Remarks to the Author):

In their manuscript entitled 'sphingosine 1-phosphate mediates adiponectin receptor signaling essential for lipid homeostasis and embryogenesis' Ruiz, Pilon and coworkers study the effect of AdipoR1/2 loss of function on membrane lipid composition in mouse embryos and derived MEFs, cell lines, and C. elegans. They find that AdipoR1/2 dKO mouse embryos and derived MEFs have increased membrane phospholipid saturation and packing, S1P administration mitigates these effects and those of 'overwhelming' feeding of WT cells with palmitic acid. The authors then go on and, through a long series of articulated experiments, propose a pathway where AdipoR-generated sphingosine is phosphorylated to S1P by SphK1 and 2, S1P then signals through S1PR3 and PPARg to activate SREBP1 and induce SCD that in turn remodel the membrane lipid composition.

While interesting in principle, the findings reported in this manuscript are not always investigated to the required degree. Several of the claims the authors make are not fully corroborated by the data which undermines the overall cogency of the study. Thus, in spite of its remarkable width this report seems, at times, to lack the proper depth.

Specifically:

1. The authors claim that "excess membrane phospholipid saturation explains AdipoR1/2 DKO embryonic lethality". They observe that AdipoR1/2 DKO develop fairly normally up to E12.5 while they are severely impaired at day 15.5. Lipidomics and proteomics at day 12.5 reveals enrichment of palmitic acid containing lipids and altered levels of proteins involved in adipogenesis in AdipoR1/2 DKO embryos. Nonetheless no information is provided on the mechanism of embryo regression/ re-absorbance and there is no solid proof for the lipid imbalance to cause lethality.

2. When looking at MEFs from AdipoR1/2 DKO embryos the authors observe an interesting phenotype possibly involving ER membranes. This is an intriguing observation but largely left unexplored. The ER is a multidomain organelle where ER resident can partition differently in the different domain. What proteins populate these ER structures induced by AdipoR1/2 DKO? Which functions of the ER are impaired in MEFs from AdipoR1/2 DKO embryos? Is membrane trafficking in and out of the ER working properly? Are lipid droplets formed properly by these cells? Are ER contacts with mitochondria and other organelles preserved?

3. The authors show that treating cells with S1P rescues several of the membrane defects induced by AdipoR1/2 loss of function in different systems. The reasoning behind the use of S1P is that "AdipoR1/2 possess an intrinsic ceramidase activity in vitro that can catalyze the hydrolysis of ceramides to produce sphingosine and free fatty acids. The resulting sphingosine can be phosphorylated to produce the signaling molecule S1P." While this reasoning is fair this hypothesis needs to be tested experimentally. Thus, levels of ceramides, sphingosine and S1P need to be measured in WT and AdipoR1/2 DKO MEF (and AdipoR2 KD HeK293) cells upon treatment with PA. Is the AdipoR1/2 ceramidase activity induced by PA treatment? If so, is the increase in membrane packing a signal for AdipoR1/2 to process more ceramide and obtain S1P? What is the source of ceramide dealt by AdipoR1/2 upon increased membrane packing conditions? What is the role of adiponectin in this process?

4. The authors then go on to manipulate cells signaling pathways downstream of S1P production by the use of inhibitors and evaluate the effects of these treatments on membrane packing. By doing so they reconstruct a pathway for the action of AdipoR1/2 whereby they conclude that "Parallel activation of S1PR3 and PPAR γ by S1P mediates AdipoR2 membrane protection" and that "Fatty acid desaturation is the final outputs of the AdipoR2-S1P signaling pathway" In order to be able to conclude this, two main things need to be done. i) The authors should provide proofs that the concerned signaling pathways are activated upon increased membrane packing; ii) genetic manipulations should match the use of inhibitors.

Reviewer #3 (Remarks to the Author):

The manuscript #NCOMMS-21-38753 entitled "SPHINGOSINE 1-PHOSPHATE MEDIATES ADIPONECTIN RECEPTOR SIGNALING ESSENTIAL FOR LIPID HOMEOSTASIS AND EMBRYOGENESIS" is addressing several important hypotheses related to the AdipoR2 signaling and saturation of fatty acids as well as the role of S1P on this axis. The paper is well written and contain novel elements but also important confirmation of previous related studies. Despite this, some parts of the authors work may still be improved to clarify the data or strengthens the conclusions.

1. Introduction could benefit from a clear hypothesis instead of a summary of results obtained
2. Page 5: it is stated “These results are consistent with AdipoR1/2 being essential proteins that sustain fatty acid desaturation to regulate membrane composition /fluidity.” This statement seems not to be supported as such by the data. It is correct that lipidomic profiles show changes in the amount of saturated/desaturated fatty acids depending on AdipoR genotype. However, the data does not contain measurements per se on desaturase activity or content. It can suggest this is a possibility, but evidence for such statement needs to be included.
3. Page 6: does the author have any explanation for why apoM and apoE should be affected by the present of adipoR2? And what impact can these observations have on the data for example related to S1P later in the paper.
4. Page 6: Fig 2A and related text “DKO MEFs showed membrane composition defects similar to those observed in embryos”. It is not clear from the figure that the difference should be there – unless the cells are stimulated with PA, but at baseline it seems unchanged between genotypes.
5. Page 7: Fig 2H-J. The legends and layout can be misinterpreted as WT/R2KO/DKO (related to E-G).
6. Page 7: “Additionally, nuclear envelope budding events, which have been previously identified in cells under several stress conditions 29, were also present in most of the DKO MEFs upon PA treatment (Figure 2D and S2L-M).” Are Fig 2D the right figure in this sentence? That does not reflect envelope budding events.
7. Page 7, Fig 2M-P: S1P treatment may reduce the amount of PA in PE but does not rescue to a WT level. What can explain this?
8. Page 8: The authors should extrapolate on why the use of c.elegance is needed for the conclusions of the paper. How is S1P levels in Paqr2 mutants? At page 10 – when addressing the use of Sphk-1/2 mutants is will again be relevant to include S1P levels.
9. Page 8 “Interestingly, WT cells reacted to PA by boosting SCD activity to convert PA into PalOA and hence prioritizing desaturation over elongation (Figure 4C-D, Figure S4F-I and Table S5).” Figure 4C-D seems to be mistakenly used – believe its should be 3C-D.
10. Page 9: The authors have included significant amount of information from other cell lines. It seems not to add much to the final conclusions. One example could be enough if at all needed.
11. Page 9: “Altogether, our results demonstrate that S1P promotes membrane fluidity and suggest that the AdipoR2 ceramidase activity is evolutionary conserved and essential for signaling through S1P”. In the present manuscript there is no measurements on ceramidase activity in assays +/-S1P in various genotypes. That would be needed for such statement.
12. Page 10: The authors have included A971432 as a S1PR5 agonist. What are the rational behind this – why not use an antagonist?
13. Page 10: Fig 5D. In general figure legends for all figures needs improvement to assure the reader can understand the data. For fig 5D – what are the color representing?

14. Page 11: Fig S5P. It is difficult to appreciate that S1P treatment affect the SREBP1 levels. Despite the use of the loading control tubulin the bar graph does not seems to reflect the WB presented.
15. The author states at page 11 “Altogether, these results indicate that AdipoR2-derived S1P acts via S1PR3 and SREBP1 to promote fatty acid desaturation and adequate membrane fluidity”. To support this statement it will be relevant to perform lipidomics on experiments before/after S1P treatment to verify whether S1P change the saturation degree.
16. Page 11: U-2 OS cells were included in the PPAR-experiments. What is the explanation for this? Could it be excluded without loss of relevant data?
17. Page 12: The author have nicely addressed the expression levels of specific desaturases – however, the protein levels are not shown.
18. The discussion does not contain considerations on limitations for the made conclusions
19. In general figure legends are not consistent and lack identification of colors, shapes ect.

Reviewer #4 (Remarks to the Author):

Ruiz et al., present a detailed study of the role of the adiponectin receptors in the maintenance of cell membrane physical properties. A mechanism is provided that indicates that AdipoR-derived S1P, via its receptor S1PR3, mediates membrane properties by coordinately activating transcription factors SREBP1 and PPAR-gamma. This pathway serves to maintain levels of the fatty acid desaturase SCD, which directly effects membrane physical properties by modulating lipid saturation/desaturation. Mouse and C. elegans animal model studies and unbiased proteomics and lipidomics assessments support the authors’ postulated mechanism.

One concern regards the interpretation of the lipidomics heat map (Fig. 1F), as being indicative of the overall lipidome shift to acyl saturation on the Kos/DKO mice. While this interpretation does seem reasonable based on the heatmap, unbiased lipidomic data should be analyzed for enrichment using an unbiased tool such as LION (Lipid ontogeny) to further determine the scope of this effect and the potential detailed impacts on membrane structure and function.

In this regard, while information is presented regarding public access of proteomics data via PRIDE, public access to the authors’ unbiased lipidomics data (e.g., Metabolomics Workbench) should be indicated, for example, to help support statements like “excess SFF in phospholipids...loss of membrane fluidity” are supported by the lipidomics data, in addition to the cited literature, as well as to allow others to investigate these interesting data.

Regarding lipidomics, this reviewer did not find instrument and acquisition parameters provided for the MS analysis in the Sciex 5500 TripleQuad. Please provide any MS acquisition workflow approach and instrument settings.

NCOMMS-21-38753

Point-by-point response to Review Comments

We thank all four reviewers for their judicious comments on the manuscript. The reviewers were generally very positive about the presented work and made useful suggestions. We have now made several new experiments, extensively reorganized the figures and modified the text to address the reviewer's comments, all while keeping to the 6000-word limit. The result is a much improved and easier-to-read manuscript. Below are the reviewer comments (black text) and our responses (red text).

Reviewer #1 (Remarks to the Author):

Ruiz, Pilon and colleagues present a very dense data set with a lot of exciting data that will prove to be quite stimulating for the field. Based on data obtained from AdipoR1/2 double knock out embryos as well as cell lines and *C. elegans* studies, the authors show that double KO's display excessive membrane phospholipid saturation. This leads them to suggest that consistent with the previously identified enzymatic activity of the receptors, the AdipoR1/2 ceramidase activity converts ceramides (generated from exogenous palmitate) to sphingosine, which in turn gets converted to sphingosine phosphate, activating the S1P receptors. They show that this triggers activation of SREBP1, that activates together with S1P-activated PPARgamma the transcription of SCD1, which in turn mediates the desaturation of membrane phospholipids.

A very interesting set of observations that lead the authors to some interesting conclusions. While the proposed model may be somewhat oversimplified, most of the conclusions seem warranted base on the data that the authors present.

1. Given the data density, the paper is rather hard to follow. More importantly, the figures are extremely complex. While the rationale that the authors used to further underline their finding with studies in *C. elegans*, it may be better to remove the *C. elegans* data into a separate future paper and focus on the results in mammalian cells.

We took to heart the reviewer's comments and have now moved all *C. elegans* panels to supplementary figures, and also simplified the main figures from 7 to 6. Including the *C. elegans* results in the present paper highlights that S1P-mediated membrane homeostasis is evolutionary conserved and will hopefully attract a wider audience.

Despite the complexity of the data presented, there is a simple take home message for the reader, namely that the main cellular role of the adiponectin receptors is fatty acid desaturation of membrane phospholipids. That is an excellent postulate, and a lot of existing data in the field on adiponectin and its receptors may be explained on the basis of this mechanism.

A few things should be clarified specifically:

2. Fig.1D needs to be better explained. "Lipidomics analysis of the membrane phospholipids phosphatidylcholine (PC) and phosphatidylethanolamine (PE), triacylglycerols (TAG), free cholesterol (FC) and the sphingolipids sphingomyelin (SM), ceramide (Cer), dihydroceramide (DiCer), glucosylceramide (GluCer) and lactosylceramide (LacCer) allowed for a clear separation of WT, AdipoR1-KO,

AdipoR2-KO and DKO E12.5 embryos in a principal component analysis (PCA)". There has to be a better way to present that data!

We have now simplified this sentence, which now reads (p. 3): "Lipidomics analysis of multiple lipid types allowed for a clear separation of WT, AdipoR1-KO, AdipoR2-KO and DKO E12.5 embryos in a principal component analysis (PCA) (Fig.1D)."

3. It is interesting that exogenous S1P can reconstitute the AdipoR deficiencies with respect to membrane fluidity. In many instances, addition of exogenous S1P can drive substrate-driven reactions the other way. Does addition of S1P lead to increased ceramides? This should be determined in one of the systems used.

We performed the experiment suggested by the reviewer and found that the addition of S1P increased Cer and GlcCer in control cells (Supplementary Fig.3L-N) but not in AdipoR2-deficient cells. Instead, in AdipoR2 siRNA treated cells, the addition of S1P led to increased SM, GlcCer and LacCer (Supplementary Fig.3M-O) and decreased DiCer (Supplementary Fig.3K), thus producing a more control-like profile.

In the text, we now write (p. 6): "AdipoR2 silencing in the presence of exogenous palmitate abnormally activates the *de novo* ceramide synthesis pathway, including increased levels of DiCer³¹. Here we found that S1P partially normalizes the levels of DiCer, SM, GlcCer and LacCer (Supplementary Fig.3K-O and Data 6). Consistently, S1P also improved membrane packing, as indicated by a lower GP index (Supplementary Fig.3P-Q)".

4. Perhaps the weakest link in their scheme relates to SREBP activation. Again, the *C. elegans* data is not that helpful, and the authors should provide additional data in the mammalian system demonstrating the nature of the downstream SREBP activation.

To strengthen the SREBP-related results, we now compared RNAseq data from HEK293 cells lacking either AdipoR2 or SREBF1/2 and treated with palmitic acid (Supplementary Fig.6E-J). Importantly, we found that over 77% of the 84 SREBF1/2-regulated genes are also AdipoR2-dependent genes and in concordant directions (up/downregulated).

In the text we now write (p. 9): "Beyond the similar membrane packing defects, the transcriptome of AdipoR1/2 and SREBP1/2 deficient cells should be similarly affected if both proteins are in the same pathway. To address this question, we compared the RNAseq profiles of HEK293 cells lacking AdipoR2² or SREBP1/2⁴¹ (Supplementary Fig.6E-J) in basal media and after PA treatment. 84 SREBP-regulated genes separated WT and AdipoR2-KO cells after PA treatment in a PCA plot (Supplementary Fig.6E). Importantly, the majority of those SREBP-regulated genes (77.4%) were also AdipoR2-regulated genes (Supplementary Fig.6F), and the concordance was greatest among the most misregulated genes: i.e., SCD and INSIG (lipid synthesis) and HERPUD1 and JUN (ER-stress) (Supplementary Fig.6G-J)".

Nevertheless, these points should not distract from the fact that this is an outstanding set of observations that significantly advance the field!

Reviewer #2 (Remarks to the Author):

In their manuscript entitled ‘sphingosine 1-phosphate mediates adiponectin receptor signaling essential for lipid homeostasis and embryogenesis’ Ruiz, Pilon and coworkers study the effect of AdipoR1/2 loss of function on membrane lipid composition in mouse embryos and derived MEFs, cell lines, and *C. elegans*. They find that AdipoR1/2 dKO mouse embryos and derived MEFs have increased membrane phospholipid saturation and packing, S1P administration mitigates these effects and those of ‘overwhelming’ feeding of WT cells with palmitic acid. The authors then go on and, through a long series of articulated experiments, propose a pathway where AdipoR-generated sphingosine is phosphorylated to S1P by SphK1 and 2, S1P then signals through S1PR3 and PPAR γ to activate SREBP1 and induce SCD that in turn remodel the membrane lipid composition. While interesting in principle, the findings reported in this manuscript are not always investigated to the required degree. Several of the claims the authors make are not fully corroborated by the data which undermines the overall cogency of the study. Thus, in spite of its remarkable width this report seems, at times, to lack the proper depth. Specifically:

1. The authors claim that “excess membrane phospholipid saturation explains AdipoR1/2 DKO embryonic lethality”. They observe that AdipoR1/2 DKO develop fairly normally up to E12.5 while they are severely impaired at day 15.5. Lipidomics and proteomics at day 12.5 reveals enrichment of palmitic acid containing lipids and altered levels of proteins involved in adipogenesis in AdipoR1/2 DKO embryos. Nonetheless no information is provided on the mechanism of embryo regression/ re-absorbance and there is no solid proof for the lipid imbalance to cause lethality.

We agree with the reviewer: we do not know the specific cause of the embryonic lethality in the DKO animals. However, the rather dramatic excess saturation in membrane phospholipids precedes embryonic failure in these animals and, given the key role of the AdipoRs in membrane homeostasis (this and previous work), we feel that a connection between the membrane defect and the embryonic lethality is a well-supported hypothesis.

On p. 4, we now more cautiously write: “Collectively, our results suggest that AdipoR1/2 DKO embryonic lethality may be caused by metabolic problems that are secondary consequences of membrane malfunction due to an excess of SFA-containing phospholipids”.

Additionally, we now mention the reviewer’s point in the discussion (p. 12): “Even though there is a pronounced lipid imbalance in DKO embryos (Fig.1D-E, I-J and Supplementary Fig.1I-J, L), a limitation of our study is that more mechanistic data are required to fully explain their regression/re-absorbance.” (...) “The present work leaves several questions open. In particular, how membrane homeostasis is regulated in adult mammals. In the future it would be interesting to attempt inducible KO mouse models or tissue specific AdipoR-KOs to identify critical tissues.”

2. When looking at MEFs from AdipoR1/2 DKO embryos the authors observe an interesting phenotype possibly involving ER membranes. This is an intriguing observation but largely left unexplored. The ER is a multidomain organelle where ER resident can partition differently in the different domain. What proteins populate these

ER structures induced by AdipoR1/2 DKO? Which functions of the ER are impaired in MEFs from AdipoR1/2 DKO embryos? Is membrane trafficking in and out of the ER working properly? Are lipid droplets formed properly by these cells? Are ER contacts with mitochondria and other organelles preserved?

The reviewer is completely correct that there is a whole avenue of research concerning the ER defects in AdipoR-deficient cells. Although a more comprehensive study is mostly beyond the scope of the present paper, we have on the reviewer's performed a more detailed analysis of the ER/mitochondria contact points and of lipid droplets formation. Specifically, using electron microscopy images, we found that ER contacts with mitochondria are preserved despite the membrane abnormalities in DKO MEFs treated with PA. This is now shown in new Supplementary Fig.2S-V. Additionally, we visualized lipid droplets and found that MEFs do not make multiple/big lipid droplets upon PA stimuli and therefore lipid droplets are hard to detect with light microscopy, though it is possible to find them with EM in a similar frequency in WT and DKO. Additionally, oleic acid, a potent inducer of lipid droplet formation, allowed us to visualize lipid droplets in both genotypes using confocal microscopy (new Supplementary Fig 3A-C). Incidentally, several evidences indicates that lipid droplet formation is not particularly altered in AdipoR-deficient cells: the ratio TAG/PC is not altered in DKO embryos at E12.5, but enriched in SFA (Supplementary Fig.1J and Data 2), as was the case in HEK293, HepG2 and 1321N1 cells that we described in previous studies (Ruiz et al 2019 JLR and Ruiz et al 2021 BBA Lipids).

In the text, we now write (p. 5): "These membrane defects are likely ER abnormalities given the fluorescent microscopy study described earlier (Fig.2E-J). Note however that DKO MEFs preserved ER-mitochondria contact points (Supplementary Fig.2S-V) and the ability to make lipid droplets (Supplementary Fig.3A-C) in basal conditions and when treated with PA or OA".

3. The authors show that treating cells with S1P rescues several of the membrane defects induced by AdipoR1/2 loss of function in different systems. The reasoning behind the use of S1P is that "AdipoR1/2 possess an intrinsic ceramidase activity in vitro that can catalyze the hydrolysis of ceramides to produce sphingosine and free fatty acids. The resulting sphingosine can be phosphorylated to produce the signaling molecule S1P." While this reasoning is fair this hypothesis needs to be tested experimentally. Thus, levels of ceramides, sphingosine and S1P need to be measured in WT and AdipoR1/2 DKO MEF (and AdipoR2 KD HeK293) cells upon treatment with PA. Is the AdipoR1/2 ceramidase activity induced by PA treatment? If so, is the increase in membrane packing a signal for AdipoR1/2 to process more ceramide and obtain S1P? What is the source of ceramide dealt by AdipoR1/2 upon increased membrane packing conditions? What is the role of adiponectin in this process?

The reviewer raises several very important questions, which we have at least partially answered with new experiments. Specifically, S1P and Sph levels in MEFs were measured to indirectly quantify ceramidase activity. We found that MEFs increased S1P production after PA 400 μ M treatment (Fig. 3A and Supplementary Data 4). We also found that S1P and Sph are reduced in mouse brains of AdipoR2-KO embryos at E15.5 and in DKO MEFs (Fig 1L, S1L, 2M-N), and that ceramides were increased in HEK293 cells treated with AdipoR2 siRNA and PA (Supplementary Fig.3L). These results are in agreement with the literature, e.g., Holland et al Nat Med 2011 found

that S1P is induced upon PA stimulation in WT MEFs, but that DKO MEFs cannot match this production. Therefore, a plausible interpretation is that an increase of membrane packing signals for AdipoR1/2 to process ceramides to obtain Sph and S1P.

In the text, we now write:

p. 4: “Indeed, the excessive saturation of PC in AdipoR2 KO embryonic brains at E15.5 (Fig.1K and Supplementary Data 2) was accompanied by a significant reduction of S1P and its precursor sphingosine (Sph) (Fig.1L and Supplementary Fig.1L). Note that the frequency of AdipoR1/2 DKO embryos at E15.5 is very low and S1P was below detection in the single AdipoR1/2 DKO embryo isolated (Fig.1A and L)”.

p. 5: “Finally, and as in the brains of the AdipoR2-KO embryos at E15.5, S1P and its precursor Sph were reduced in DKO MEFs (Fig.2M-N and Supplementary Data 4) under basal conditions, or after PA treatment”.

p.6: “AdipoR2 silencing in the presence of exogenous palmitate abnormally activates the *de novo* ceramide synthesis pathway, including increased levels of DiCer³¹. Here we found that S1P partially normalizes the levels of DiCer, SM, GlcCer and LacCer (Supplementary Fig.3K-O and Data 6)”.

p. 6: “Small amounts of PA are particularly toxic to cells/organism lacking AdipoR1/2^{10, 24}. Even though WT cells react to high doses of PA by increasing S1P and Sph production (Fig.3A and Supplementary Data 4), WT cells are also sensitive to high concentrations of PA and suffer membrane rigidification and death^{14, 33}.”

More data would be needed to confidently describe the source of the ceramide processed by the AdipoRs. It could be a combination of different sources: (i) ceramide already present in the plasma membrane, (ii) *de novo* synthesis and (iii) degradation of more complex sphingolipids such as sphingomyelin. The use of inhibitors/siRNA against the different enzymes involved in sphingolipid metabolism and labelled substrates could help to identify the source(s) of the ceramide process by the AdipoRs.

Our cell culture experiments were mostly carried out in serum free media, i.e., Adiponectin-free media. Also, Adiponectin did not have any effect on membrane fluidity when exogenously added in a previous study (Ruiz et al. 2019, JLR). It is possible that the “right form” of Adiponectin (i.e., specific configuration/structure/modification) could modulate AdipoR1/2 ceramidase activity as previously shown (Holland et al 2011 Nat Med). Nevertheless, from our present and previous work it seems clear that AdipoR1/2 can regulate membrane fluidity independently of Adiponectin.

4. The authors then go on to manipulate cells signaling pathways downstream of S1P production by the use of inhibitors and evaluate the effects of these treatments on membrane packing. By doing so they reconstruct a pathway for the action of AdipoR1/2 whereby they conclude that “Parallel activation of S1PR3 and PPAR γ by S1P mediates AdipoR2 membrane protection” and that “Fatty acid desaturation is the final outputs of the AdipoR2-S1P signaling pathway” In order to be able to conclude this, two main things need to be done. i) The authors should provide proofs that the

concerned signaling pathways are activated upon increased membrane packing; ii) genetic manipulations should match the use of inhibitors.

In essence, the reviewer asks for additional support for the proposed AdipoR-S1P-S1PR3-SREBP/PPAR γ -SCD pathway. We have now performed additional experiments/analyses along the lines suggested by the reviewer.

i) As mentioned in response to Reviewer #1, point 4, we have now found that the same genes are misregulated in HEK293 cells lacking AdipoR2 or SREBP1/2 and after PA treatment (increased membrane packing situation). See the new Supplementary Fig.6E-J. Additionally, SREBF1/2 transcription is downregulated in AdipoR2-KO HEK293 cells treated with PA (Supplementary Fig.6F). We also found that silencing PPARG also decreased SCD at protein level (Fig.6B) in HEK293 cells treated with PA, and this pathway appears conserved in *C. elegans* where S1P can substitute for PAQR-2 to promote nucleolar localization of the PPAR homolog NHR-49 (Supplementary Fig. 7Q-V).

ii) In the manuscript, we systematically show that the effect of genetic manipulations (through RNAi) matches the use of inhibitors for the three targets tested: SREBF1 siRNA and betulin, PPARG siRNA and T0070907, S1PR3 siRNA and TY52156 (Fig.4-5 and Supplementary Fig.5-7).

We also verified that SCD protein levels are reduced in the cells lacking AdipoRs or PPARG, and now write (p. 10, bottom): "Here, we confirmed that DKO MEFs and HEK293 cells lacking AdipoR2 or PPAR γ have reduced SCD at protein and mRNA levels (Fig.6A-B and Supplementary Fig. 8A-B)".

Reviewer #3 (Remarks to the Author):

The manuscript #NCOMMS-21-38753 entitled " Sphingosine 1-Phosphate Mediates Adiponectin Receptor Signaling Essential for Lipid Homeostasis and Embryogenesis" is addressing several important hypotheses related to the AdipoR2 signaling and saturation of fatty acids as well as the role of S1P on this axis. The paper is well written and contain novel elements but also important confirmation of previous related studies. Despite this, some parts of the authors work may still be improved to clarify the data or strengthens the conclusions.

1. Introduction could benefit from a clear hypothesis instead of a summary of results obtained.

We have modified the introduction and now state a hypothesis as starting point (p. 4, end of introduction): "Here, to better understand the cause of AdipoR1/2 embryonal lethality, we generated DKO mouse embryos and explored the hypothesis that deficient S1P signaling may have led to a failure in membrane homeostasis."

2. Page 5: it is stated "These results are consistent with AdipoR1/2 being essential proteins that sustain fatty acid desaturation to regulate membrane composition /fluidity." This statement seems not to be supported as such by the data. It is correct that lipidomic profiles show changes in the amount of saturated/desaturated fatty acids depending on AdipoR genotype. However, the data does not contain measurements

per see on desaturase activity or content. It can suggest this is a possibility, but evidence for such statement needs to be included.

Western-Blots showing a reduction of SCD in DKO MEFs and in AdipoR2-lacking cells are now included (Fig.6A-B). Also, SCD and FADS2 expression/protein levels are reduced in HUVEC and HEK293 cells lacking AdipoR2 (Ruiz et al. 2019 JLR and 2022 BBA Lipids). Identical results were obtained in the model organism *C. elegans*: lower levels of the desaturase fat-7 in the *paqr-2* null mutant (Supplementary Fig.8 C-D). Importantly, and supporting the model presented in this manuscript, S1P boosted fat-7 levels (Supplementary Fig.8J-K).

In the text, we now write (p. 10, bottom): “Remarkably, an RNAseq experiment previously showed that SCD is the most robustly downregulated gene in AdipoR2-KO HEK293 cells ². Here, we confirmed that DKO MEFs and HEK293 cells lacking AdipoR2 or PPAR γ have reduced SCD at the protein and mRNA levels (Fig.6A-B and Supplementary Fig. 8A-B). Likewise, in *C. elegans*, it has been shown that *paqr-2* mutant worms have reduced expression of the SCD ortholog *fat-7* ¹¹. Here, we confirmed that *fat-7* is downregulated in *paqr-2* mutant worms (Supplementary Fig.8C-E) and quantified the expression of the other two Δ -9 desaturases in *C. elegans*, namely *fat-5* and *fat-6*, and found that these were also downregulated in the *paqr-2* mutant (Supplementary Fig.8F-G).”

3. Page 6: does the author have any explanation for why apoM and apoE should be affected by the present of adipoR2? And what impact can these observations have on the data for example related to S1P later in the paper.

We do not have an explanation for how AdipoR1/2 deficiency leads to the observed decreased levels of apolipoproteins. However, these reduced levels could have important organism-wide effects on membrane homeostasis, as we now write in the text (p. 12): “Limited availability of ApoM, an S1P carrier in plasma, likely hampers S1P transport in the embryo. ApoM expression in mouse embryos is strongly induced at E11 ⁴⁹, suggesting that S1P transport is important from this stage. Membrane homeostasis can be regulated cell nonautonomously ²⁵, and S1P could be acting as a signal molecule between tissues to globally maintain membrane homeostasis.”

4. Page 6: Fig 2A and related text “DKO MEFs showed membrane composition defects similar to those observed in embryos”. It is not clear from the figure that the difference should be there – unless the cells are stimulated with PA, but at baseline it seems unchanged between genotypes.

We agree with the reviewer that the lipid composition of DKO MEFs in basal media is not deeply affected. However, important changes can be identified, i.e., an increase of the SFA 16:0 and a decrease of the PUFA 20:4 in PE (Supplementary Fig.2A and 2C and Data 4). This is in line with the results previously obtained in several cell lines (Ruiz et al, 2019 JLR) and in *C. elegans* (Devkota et al. 2017 PLoS Genetics). One explanation could be that DKO MEFs were cultured in media containing regular serum (including lipoproteins), whereas DKO embryos have reduced levels of the main lipoproteins (Fig1I-J).

5. Page 7: Fig 2H-J. The legends and layout can be misinterpreted as WT/R2KO/DKO (related to E-G).

Thank you for noticing this. Now Fig.2H-J show R2KO.

6. Page 7: “Additionally, nuclear envelope budding events, which have been previously identified in cells under several stress conditions 29, were also present in most of the DKO MEFs upon PA treatment (Figure 2D and S2L-M).” Are Fig 2D the right figure in this sentence? That does not reflect envelope budding events.

Thank you for pointing it out. This is now corrected: Fig.2L and Supplementary Fig.2Q-R in the revised version.

7. Page 7, Fig 2M-P: S1P treatment may reduce the amount of PA in PE but does not rescue to a WT level. What can explain this?

This probably reflects our inability to supply S1P at optimal levels. In our experiments, the DKO MEFs received a single dose of S1P and are then incubated for 18 h, whereas WT cells may dynamically produce S1P as required.

8. Page 8: The authors should extrapolate on why the use of *C.elegans* is needed for the conclusions of the paper. How is S1P levels in *Paqr2* mutants? At page 10 – when addressing the use of *Sphk-1/2* mutants is will again be relevant to include S1P levels.

We have now moved all *C. elegans* panels to supplementary figures. We think that including the *C. elegans* results in the present paper highlights that S1P-mediated membrane homeostasis is evolutionary conserved and will attract a wider audience. We also agree with the reviewer that it would be relevant to measure the levels of S1P in *paqr-2* and *sphk-1* mutants. However, to our knowledge, S1P has not been measured in worms due to technical difficulties. As an alternative, we have now measured the levels of S1P and Sph in mouse embryos MEFs and HEK293 cells. R2KO embryos at E15.5 and DKO MEFs have reduced levels of S1P (Fig.1L and 2M).

9. Page 8 “Interestingly, WT cells reacted to PA by boosting SCD activity to convert PA into PalOA and hence prioritizing desaturation over elongation (Figure 4C-D, Figure S4F-I and Table S5).” Figure 4C-D seems to be mistakenly used – believe it should be 3C-D.

This has been corrected in the revised version (Fig.3D-E).

10. Page 9: The authors have included significant amount of information from other cell lines. It seems not to add much to the final conclusions. One example could be enough if at all needed.

We feel that the use of different models and cell lines strengthens the conclusions of the manuscript and shows the “universal” importance of membrane fluidity regulation. To improve the readability of the main manuscript, much of the data on other models (e.g., all *C. elegans* data, and most data on other cell lines) is provided in supplementary figures.

11. Page 9: “Altogether, our results demonstrate that S1P promotes membrane fluidity and suggest that the AdipoR2 ceramidase activity is evolutionary conserved and essential for signaling through S1P”. In the present manuscript there is no measurements on ceramidase activity in assays +/-S1P in various genotypes. That would be needed for such statement.

We have now more cautiously rephrased this sentence (p. 8, top): “Altogether, our results demonstrate that S1P promotes membrane fluidity and suggest that the conserved AdipoR2 catalytic site, capable of a ceramidase activity that would lead to S1P production^{13, 14}, is essential for membrane homeostasis.”

It is correct that we have not monitored directly the AdipoR1/2 ceramidase activity. Instead, we relied on previous studies showing that AdipoR2 has an intrinsic ceramidase activity (i.e., Vasiliauskaite-Brooks, I. et al. 2017, Nature) and studied whether S1P could functionally replace AdipoR2 in different models/experimental conditions (it does). Additionally, the new S1P measurements show that R2KO embryos at E15.5 and DKO MEFs have reduced levels of S1P (Fig.1L and 2M), which is consistent with reduced ceramidase activity in the AdipoR-deficient cells.

12. Page 10: The authors have included A971432 as a S1PR5 agonist. What are the rational behind this – why not use an antagonist?

At the time of the experiments, we did not find any S1PR5 specific antagonist commercially available. However, note that S1PR5 expression was not detected in MEFs (Fig.4F) indicating that S1PR5 is not fundamental for membrane fluidity regulation. Nevertheless, we use the agonist A971432 in HEK293 cells (a S1PR5 positive cell type, Supplementary Fig.5J) and found no changes in membrane packing.

13. Page 10: Fig 5D. In general figure legends for all figures needs improvement to assure the reader can understand the data. For fig 5D – what are the color representing?

We have gone through the figures and legends to improve their readability and adapt them to journal format guidelines (i.e., 350-word limits for legends). In most figures, black is used for control conditions, red is used to highlight rigid situations (i.e., AdipoR2 siRNA + PA) and light green to represent S1P treatments.

4. Page 11: Fig S5P. It is difficult to appreciate that S1P treatment affect the SREBP1 levels. Despite the use of the loading control tubulin the bar graph does not seems to reflect the WB presented.

While this anti-SREBP1 antibody is commonly used, it is not the cleanest one and recognized several background bands. Thus, we cautiously limited our analysis to the unprocessed form of SREBP1 and included SREBP1 siRNA treated cells as control. The bands shown in Fig.5C and S6D are a representative experiment. The bar graph represents the mean \pm SEM of three independent experiments. To strengthen the SREBP-related results, as mentioned earlier (point 2 by this reviewer), we have now compared RNAseq data of HEK293 cells lacking AdipoR2 or SREBF1/2 and treated with palmitic acid (Supplementary Fig.6E-J) and found that over 77% of SREBF1/2-regulated genes are also AdipoR2-regulated genes.

15. The author states at page 11 “Altogether, these results indicate that AdipoR2-derived S1P acts via S1PR3 and SREBP1 to promote fatty acid desaturation and adequate membrane fluidity”. To support this statement it will be relevant to perform lipidomics on experiments before/after S1P treatment to verify whether S1P change the saturation degree.

The suggested lipidomics by the reviewer are now included in Fig.6E-F, Supplementary Fig.8N and Data 6. S1P reduced the degree of saturation in HEK293 cells lacking AdipoR2, but not in S1PR3 or SREBF1 deficient cells what it is in agreement with the conclusions of the manuscript.

16. Page 11: U-2 OS cells were included in the PPAR-experiments. What is the explanation for this? Could it be excluded without loss of relevant data?

The use of the U-2 OS and other cell lines strengthens the conclusions. It shows that the results are not limited to a particular line with specific mutations/expression profile.

17. Page 12: The author have nicely addressed the expression levels of specific desaturases – however, the protein levels are not shown.

New Western-blot in Fig.6A-B show reduced levels of SCD in DKO MEFs and AdipoR2 siRNA treated cells. In addition, a knockdown in PPAR γ caused reduced expression of SCD. These results are in line with Ruiz et al. 2021 BBA: SCD and FADS2 proteins were reduced in AdipoR2-KO HEK293 cells treated with palmitic acid.

18. The discussion does not contain considerations on limitations for the made conclusions

We now write (p. 12):

“Even though there is a pronounced lipid imbalance in DKO embryos (Fig.1D-E, I-J and Supplementary Fig.1I-J, L), a limitation of our study is that more mechanistic data are required to fully explain their regression/re-absorbance.” (...) “The present work leaves several questions open and has the limitation that it relied heavily on in vitro experiments. In the future it would be interesting to address the question of systemic membrane homeostasis with inducible KO mouse models or tissue-specific AdipoR-KOs to identify critical tissues. Additionally, the mechanism of membrane fluidity sensing by AdipoR2 was not addressed in the present work. The ER-resident protein RNF145, a ubiquitin ligase, promotes AdipoR2 degradation only under high membrane fluidity conditions, and this likely contributes to its regulation (Volkmar et al. EMBO J. 2022). It would be interesting in the future to examine S1P signaling in animals where the expression of RNF145 is experimentally manipulated.”

19. In general figure legends are not consistent and lack identification of colors, shapes ect.

We are constrained by the maximum of 350 words allowed for each legend. The figures themselves now include additional labels that provide information (cell type, treatments, etc.) necessary to understand the experiments presented.

Reviewer #4 (Remarks to the Author):

Ruiz et al., present a detailed study of the role of the adiponectin receptors in the maintenance of cell membrane physical properties. A mechanism is provided that indicates that AdipoR-derived S1P, via its receptor S1PR3, mediates membrane properties by coordinately activating transcription factors SREBP1 and PPAR-gamma. This pathway serves to maintain levels of the fatty acid desaturase SCD, which directly effects membrane physical properties by modulating lipid saturation/desaturation. Mouse and C. elegans animal model studies and unbiased proteomics and lipidomics assessments support the authors' postulated mechanism. One concern regards the interpretation of the lipidomics heat map (Fig. 1F), as being indicative of the overall lipidome shift to acyl saturation on the Kos/DKO mice. While this interpretation does seem reasonable based on the heatmap, unbiased lipidomic data should be analyzed for enrichment using an unbiased tool such as LION (Lipid ontology) to further determine the scope of this effect and the potential detailed impacts on membrane structure and function. In this regard, while information is presented regarding public access of proteomics data via PRIDE, public access to the authors' unbiased lipidomics data (e.g., Metabolomics Workbench) should be indicated, for example, to help support statements like "excess SFF in phospholipids...loss of membrane fluidity" are supported by the lipidomics data, in addition to the cited literature, as well as to allow others to investigate these interesting data. Regarding lipidomics, this reviewer did not find instrument and acquisition parameters provided for the MS analysis in the Sciex 5500 TripleQuad. Please provide any MS acquisition workflow approach and instrument settings.

We thank the reviewer for the comments and realize that we have been a bit unclear in the manuscript regarding the lipidomics analysis. The lipidomics was done using a targeted approach on several platforms (both UPLC-MS/MS and direct infusion (shotgun) MS). Even though, the shotgun analysis using precursor ion and neutral loss scanning (sometimes called lipid class profiling) gives more information than the UPLC-MS/MS analysis, the use of LipidView for extraction of predefined lipids makes us consider this data set to also have been generated in a targeted approach. We have now clarified this in the manuscript (p. 15-16) and added more information about lipid analysis. Despite having data generated by targeted lipidomics we agree with the reviewer regarding data availability and have now uploaded the data set in the Zenodo database (10.5281/zenodo.7024516).

REVIEWERS' COMMENTS

Reviewer #1 (Remarks to the Author):

The authors have appropriately addressed my initial concerns and have done an excellent job with the revision of this manuscript.

Reviewer #2 (Remarks to the Author):

The authors sufficiently addressed my concerns.

Reviewer #3 (Remarks to the Author):

No further comments. The Authors have nicely answered the raised questions and considerations in the initial revision. New experiments are well incorporated and discussed. The overall quality of the revised paper is high and appropriate for the conclusions.

Reviewer #4 (Remarks to the Author):

The authors have been very responsive to every critique point, and have done an extensive and comprehensive revision, including new experiments, which make the paper much more impactful and approachable.

NCOMMS-21-38753

Point-by-point response to Review Comments

We thank all four reviewers for their judicious comments during the evaluation process. All reviewers were satisfied with the previous version and therefore no reviewer comments needed to be addressed.

REVIEWERS' COMMENTS

Reviewer #1 (Remarks to the Author):

The authors have appropriately addressed my initial concerns and have done an excellent job with the revision of this manuscript.

Reviewer #2 (Remarks to the Author):

The authors sufficiently addressed my concerns.

Reviewer #3 (Remarks to the Author):

No further comments. The Authors have nicely answered the raised questions and considerations in the initial revision. New experiments are well incorporated and discussed. The overall quality of the revised paper is high and appropriate for the conclusions.

Reviewer #4 (Remarks to the Author):

The authors have been very responsive to every critique point, and have done an extensive and comprehensive revision, including new experiments, which make the paper much more impactful and approachable.